# Analysis of the Chemical Composition and Evaluation of the Antioxidant, Antimicrobial, Anticoagulant, and Antidiabetic Properties of *Pistacia lentiscus* from Boulemane as a Natural Nutraceutical Preservative

**DOI:** 10.3390/biomedicines11092372

**Published:** 2023-08-24

**Authors:** Aziz Drioiche, Atika Ailli, Firdaous Remok, Soukaina Saidi, Aman Allah Gourich, Ayoub Asbabou, Omkulthom Al Kamaly, Asmaa Saleh, Mohamed Bouhrim, Redouane Tarik, Amale Kchibale, Touriya Zair

**Affiliations:** 1Research Team of Chemistry of Bioactive Molecules and the Environment, Laboratory of Innovative Materials and Biotechnology of Natural Resources, Faculty of Sciences, Moulay Ismaïl University, B.P. 11201, Zitoune, Meknes 50070, Morocco; a.ailli@umi.ac.ma (A.A.); f.remok@edu.umi.ac.ma (F.R.); soukainasaidi.ss48@gmail.com (S.S.); gourich.amanallah@gmail.com (A.A.G.); ayouboxy@yahoo.fr (A.A.); tarikredouane77@yahoo.fr (R.T.); amal_cer@yahoo.fr (A.K.); 2Medical Microbiology Laboratory, Mohamed V. Hospital, Meknes 50000, Morocco; 3Department of Pharmaceutical Sciences, College of Pharmacy, Princess Nourah Bint Abdulrahman University, P.O. Box 84428, Riyadh 11671, Saudi Arabia; omalkmali@pnu.edu.sa (O.A.K.); asali@pnu.edu.sa (A.S.); 4Team of Functional and Pathological Biology, Laboratory of Biological Engineering, Faculty of Sciences and Technology Beni Mellal, University Sultan Moulay Slimane, Beni Mellal 23000, Morocco; mohamed.bouhrim@gmail.com

**Keywords:** *Pistacia lentiscus*, germacrene D, spathulenol, 3,5-di-*O*-galloyl quinic acid, gallic acid, 3,4,5-tri-*O*-galloyl quinic acid, antioxydant, antimicrobien, anticoagulant, antidiabetic, preservative agent

## Abstract

*Pistacia lentiscus* L. has traditionally been employed as a diuretic and stimulant in the treatment of hypertension. Our interest centered on analyzing the chemical profile of the plant’s leaves and its in vitro, in vivo, and in silico antioxidant, antimicrobial, anticoagulant, and antidiabetic effects in order to valorize this species and prepare new high-value products that can be used in the agro-food and pharmaceutical industries. When this species’ essential oil was hydrodistilled and subjected to GC-MS analysis, the results showed that the principal components were germacrene D (17.54%), spathulenol (17.38%), bicyclogermacrene (12.52%), and terpinen-4-ol (9.95%). The extraction of phenolic compounds was carried out by decoction and Soxhlet. The determination of total polyphenols, flavonoids, and tannins of aqueous and organic extracts by spectrophotometric methods demonstrated the richness of this species in phenolic compounds. Chromatographic analysis by HPLC/UV-ESI-MS of the aqueous extract of *P. lentiscus* revealed the presence of 3,5-di-*O*-galloyl quinic acid, gallic acid, and 3,4,5-tri-*O*-galloyl quinic acid specific to this species. The study of antioxidant activity by three methods (DPPH, FRAP, and Total Antioxidant Capacity) revealed that *P. lentiscus* is a very promising source of natural antioxidants. The antimicrobial activity of the essential oil and aqueous extract (E_0_) was studied by microdilution on the microplate. The results revealed the effectiveness of the aqueous extract compared to the essential oil against Gram-negative bacteria (*K. pneumoniae*, *A. baumannii*, *E. aerogenes*, *E. cloacae*, *P. fluorescence*, *Salmonella* sp., *Shigella* sp., and *Y. enterolitica*) and candidoses (*C. krusei* and *C. albicans*). The measurements of prothrombin time (PT) and activated partial thromboplastin time (aPTT) of the aqueous extract (E_0_) can significantly prolong these tests from concentrations of 2.875 and 5.750 mg/mL, respectively. The antihyperglycemic effect of the aqueous extract (E_0_) showed a strong in vitro inhibitory activity of α-amylase and α-glucosidase compared to acarbose. Thus, it significantly inhibited postprandial hyperglycemia in Wistar albino rats. The in-silico study of the major compounds of the essential oil and extract (E_0_) carried out using PASS, SwissADME, pkCSM, and molecular docking tools confirmed our in vitro and in vivo results. The studied compounds showed a strong ability to be absorbed by the gastrointestinal tract and to passively diffuse through the blood-brain barrier, a similarity to drugs, and water solubility. Molecular docking experiments deduced the probable mode of action of the identified compounds on their respective target proteins, such as NADPH oxidase, thrombin, α-amylase, and α-glucosidase. Furthermore, given the demonstrated antioxidant, antimicrobial, anticoagulant, and antidiabetic effects, we can affirm the richness of *P. lentiscus* in bioactive molecules and its use in traditional medicine as a source of preservative agent.

## 1. Introduction

Since the start of the industrial revolution, sodium metabisulfite [1], sodium benzoate [2], butylated hydroxyanisole (BHA) [3], potassium sorbate [4], butylated hydroxytoluene (BHT) [5], sodium nitrite [6], etc., have ensured food safety and quality by inhibiting and preventing the growth of perishable and disease-causing cells and delaying the oxidation of lipids and proteins.

The most recent FactMR research (FACT2250MR) indicates that between 2023 and 2033, the world’s consumption of food preservatives is anticipated to expand at a compound annual growth rate of 4.6%. The market for food preservatives is estimated to be worth USD 3.2 billion globally in 2023 and is projected to grow to USD 5 billion by 2033. Due to their possible impacts on human health, synthetic chemical preservatives, which are widely employed in the food, pharmaceutical, and cosmetic sectors, are causing growing worry. According to studies, some of these preservatives may cause allergic reactions, hormone disturbances, reproductive hazards, skin problems, and a higher risk of developing cancer, among other possible health problems [7,8,9].

Growing evidence of the possible toxicity of these preservatives is making consumers more suspicious and calling for healthier diets [10]. Due to the possible risk posed by synthetic chemical preservatives in food, a nationwide survey in Australia found that 45% of Australians are concerned about the safety and quality of the food they purchase [11]. Plant-based preservatives are generally regarded as safe (GRAS), include bioactive chemicals that are crucial for food preservation, and have no known side effects.

Several bioactive compounds naturally present in plants have antioxidant and antimicrobial properties and can play a key role in food preservation. Bioactive compounds from plants are generally classified as phenolic compounds and are found in various plants, fruits, herbs, and spices [12]. Many studies have been reported on plant polyphenols as natural preservatives. These include rosemary [13], blueberry [14], green tea [15], thyme [16], and Moringa oleifera leaf [17]. In their review article, Das and colleagues [18] observed that despite the availability of many bioactive molecules of natural origin, including those from animals and microorganisms, the main focus in producing healthy foods is on the use of plant extracts containing a wide variety of bioactive molecules.

Significant evidence has accumulated, highlighting the crucial role of reactive oxygen species (ROS) and other oxidizing agents in the onset of many health problems and diseases [19]. Scientists have been interested in this evidence highlighting the importance of antioxidants in disease prevention, treatment, and maintaining human health. The presence of a natural antioxidant mechanism in the human body ensures various biological functions such as anti-mutagenic, anti-aging, and anti-carcinogenic responses. The preservation of health and well-being depends on these properties. Free radicals must first be stabilized or deactivated by antioxidants in order to prevent harm to vital cellular components. These properties have generated increasing interest in natural antioxidants, which are now widely used in the pharmaceutical, cosmetic, and food sectors. These substances have advantageous properties that have a significant potential for reestablishing essential balances [20].

The genus *Pistacia* belongs to the family Anacardiaceae and comprises at least eleven species [21]. The genus includes *Pistacia lentiscus* L., which has long been used in conventional medicine to treat a variety of diseases. The various components of it have been noted for their diverse chemical compositions. The species *Pistacia lentiscus*, also known as the mastic tree, is a small tree or shrub [22]. It is indigenous to the Mediterranean region and primarily grows in arid regions, maquis, and coastal locations [23]. The mastic tree is an evergreen shrub that can reach up to 4 m in height. It has fissured bark, a small, twisted stem, and persistent leaves that are fragrant, leathery, and dark green. The mastic tree’s flowers are small and yellowish-green, while the fruits are small drupes ranging from dark red to black and containing a single seed. The mastic tree is known for its resin, called mastic, obtained from cuts on the tree bark [24,25]. This resin is used in the food industry to flavor foods and beverages, as well as in the production of cosmetics, varnishes, and other products. The leaves and fruits of the mastic tree have also been used in traditional medicine for their diuretic, antispasmodic, and antiseptic properties [26].

Scientific research suggests that *Pistacia lentiscus* may help reduce oxidative stress in the body. Oxidative stress occurs when free radicals, which are naturally produced unstable molecules in the body, damage cells and tissues, causing a chain reaction that can lead to cellular damage and diseases [27]. The antioxidant properties of *Pistacia lentiscus* are due to the presence of polyphenols in its leaves, fruits, and resin. *Pistacia lentiscus* extracts have been proven to improve the cardiovascular and immunological systems as well as lower levels of oxidative stress in the lungs, kidneys, liver, and brain in vitro and in animal experiments [28]. However, further research is needed to confirm the effects of *P. lentiscus* on oxidative stress in humans, as well as to determine effective doses and appropriate administration routes.

Therefore, our study focused on the thorough phytochemical analysis of the essential oil (EO) and extracts of *P. lentiscus* from the Boulemane region. Additionally, the objectives of our study included identifying natural antioxidant chemical substances and evaluating the in vitro, in vivo, and in silico antioxidant, antibacterial, anticoagulant, and antidiabetic properties of the extracts and EO of this species.

## 2. Materials and Methods

### 2.1. Vegetal Material

The *P. lentiscus* sample that was studied was taken from cultivated stands in the Boulemane area (Figure 1). Table 1 displays information on the place of harvesting, the component that was harvested, and the origin. This species was identified by Pr. Hamid Khamar at the Laboratory of Botany and Plant Ecology of the Scientific Institute of Rabat.

### 2.2. Microbial Materials

Twenty-four bacterial and eight fungal strains were used to determine the antimicrobial capacity of EO and the aqueous extract of *P. lentiscus* (Table 2). These specific bacteria are dangerous and well-known for their toxic nature toward people, great resistance, and capacity for invasion. Infections are a common occurrence in Morocco, and they provide a problem in medicine and therapy. The setting for the identification of these strains was the Mohamed V-Meknes Provincial Hospital. All strains were revived in Mueller–Hinton and Sabouraud broths, subcultured, and stored in a 20% glycerol stock at 80 °C.

### 2.3. Animal Selection for Research

Albino mice, both male and female, were employed in the acute toxicity study. Wistar rats (males and females) were used in the testing of the antidiabetic efficacy in vivo. The animals were kept in the animal home for the biology department at the faculty of sciences in Oujda, which had a temperature of 22 ± 2 °C and a photoperiod of 12 h of light and 12 h of darkness. The animals were kept in good circumstances for upbringing with unrestricted access to food and water. The institutional animal ethics committee’s requirements and the care, usage, and handling of the animals were strictly followed (02/2019/LBEAS) [29].

### 2.4. Quality Control of Plant Material

Quality control of plant material is essential to ensure its purity, which involves performing precise measurements on various characteristic parameters. In this study, quality control analyses were carried out on *P. lentiscus* leaves, including moisture, pH, ash content, and elemental impurities (ICP). The approach chosen for measuring moisture adhered to the AFNOR standard (NF-V03-402 1985) [30]. The method described by Ailli et al. was used to determine the pH of *P. lentiscus* powder [22]. The ash content was determined in accordance with the standard (NFV05-113, 1972) [31]. As for the search for heavy metals, the concentrations of arsenic (As), lead (Pb), chromium (Cr), antimony (Sb), iron (Fe), cadmium (Cd), and titanium (Ti) were evaluated using an inductively coupled plasma atomic emission spectrometer (ICP-AES) of Ultima2 Jobin Yvon model, within the laboratory of the UATRS (Technical Support Unit for Scientific Research) of CNRST in Rabat [32].

### 2.5. The Qualitative and Quantitative Study of Essential Oils

#### 2.5.1. Extraction of Essential Oils from *P. lentiscus* and Determination of Yield

Using a Clevenger-style apparatus, hydrodistillation was used to extract the essential oils (EOs) from *P. lentiscus* leaves. The process involved boiling 20 g of plant material in 200 mL of distilled water for three hours and then drying the EOs with anhydrous sodium sulfate (Na_2_SO_4_). The EOs were stored in a dark bottle at 4 °C until needed. The yield of EO was calculated using Equation (1) based on the 20 g of plant material used [33].
(1)$$Yield(\%)=\frac{W(EO)}{W0}\times 100$$
where: W (EO): weight of HE recovered (g); W0: weight of plant material (20 g).

#### 2.5.2. Analysis and Identification of the Chemical Composition of *P. lentiscus* EO

The sample of the investigated EO was put through chromatographic examination utilizing a Thermo Electron (Trace GC Ultra, Milan, Italy) gas chromatograph and a Thermo Electron Trace MS system mass spectrometer (Thermo Electron: Trace GC Ultra; Polaris Q MS). Using an electron collision with a 70 eV energy level, fragmentation was achieved. A flame ionization detector (FID) and DB-5 type column (5% phenyl-methyl-siloxane; 30 m 0.25 mm 0.25 m film thickness) were both installed in the chromatograph. The column temperature was programmed to rise from 50 to 200 °C at a rate of 4 °C/min for 5 min. Nitrogen was used as the carrier gas, flowing at a rate of 1 mL/min in split injection mode (leak ratio: 1/70).

By calculating and comparing the compounds’ Kovats indices (KI) with those of the well-known standard products listed in the databases of Kovats [34], Adams [35], and Hübschmann [36], the chemical composition of the EO was identified.

To identify each molecule, the retention durations of the peaks were compared to authentic standards known to be held in the authors’ lab. Additionally, their claimed KI and MS data were matched to standards held in the WILEY and NIST 14 standard mass spectrum database and published literature. The Kovats index was utilized for this. Kovats indices were used to compare the retention times of any products to the retention times of linear alkanes with the same amount of carbons. They were established by concurrently injecting a conventional C_7_–C_40_ alkane combination under identical working conditions.

### 2.6. Phytochemical Screening

The identification of chemical families in a qualitative study of *P. lentiscus* leaves involved conducting tests for compound solubility, precipitation, and turbidity responses. Additionally, the identification process included examining samples under UV light or observing for a certain color shift. The study also involved phytochemical analysis, which was conducted by grinding dry plant samples into a fine powder and characterizing several chemical groups using methods described by Dohou et al., Judith, Mezzoug et al., Bekro et al., Bruneton, and N’Guessan et al. [37,38,39,40,41,42].

### 2.7. Study of Phenolic Compounds

#### 2.7.1. Extraction of Phenolic Compounds

To extract phenolic chemicals, both decoction and solid-liquid extraction by the Soxhlet device were employed. The decoction procedure involved adding 30 g of the sample to 600 mL of distilled water, heating it, and then boiling it for an hour at 80 °C. After settling for five minutes, the mixture was filtered at a lower pressure, and the decocted extract was recovered as a powder in a glass vial. The extract was placed in storage after being dried in a 70 °C oven. The Soxhlet apparatus was used to extract the second and third samples, each weighing 30 g, in the solid–liquid extraction method. The extraction solvent was either 300 mL of water or a 70/30 ethanol/water solution. The extracts were concentrated using a rotary evaporator following several extraction cycles. The extracts were coded as described in Table 3.

#### 2.7.2. Determination of Total Polyphenols

To determine the total polyphenol content of the study’s plant extracts, the Folin-Ciocalteu method was used, as described by Singleton and Rossi [43]. This method involves the oxidation of polyphenolic compounds and the emergence of color, and it is the most commonly used technique for this purpose. The Folin-Ciocalteu reagent, which involves reducing phosphotungstic (H_3_PW_12_O_40_) and phosphomolybdic (H_3_PMo_12_O_40_) acids, produces a combination of tungsten (W_8_O_23_) and molybdenum (Mo_8_O_3_) blue oxides. The chemicals are assayed using colorimetry and an optical density reading. The absorbance reading was obtained using a UV mini-1240 spectrophotometer with the 760 nm setting and compared to a blank (a reaction mixture without extract). A parallel calibration curve was created under the same operating conditions using a concentration range of 50 μg/mL of gallic acid as a positive control. The results were computed using the type Y = ax + b equation that was found using the calibration curve, and the results were reported as the milligram equivalent of gallic acid per gram of extract (mg GAE/g). There were three runs of each test.

#### 2.7.3. Determination of Flavonoids

To measure the flavonoid content, the colorimetric method and aluminum trichloride were used, following the techniques developed by Djeridane [44] and Hung [45] and their colleagues. Aluminum chloride (AlCl_3_) reacts with the hydroxyl groups (OH) of flavonoids to produce a compound, which was then calculated using UV spectroscopy at a wavelength of 433 nm. A calibration curve (type Y = ax + b) was created using quercetin, a standard flavonoid with concentrations between 5 and 30 μg/mL, under the same analytical conditions as the samples. The number of flavonoids is expressed in milligrams of quercetin equivalent per gram of extract (mg EQ/g). Each test was run three times.

#### 2.7.4. Determination of Condensed Tannins

The vanillin technique was used to assess the condensed tannin concentrations [46]. In this method, a vanillin/methanol solution (4%; *m*/*v*) was mixed with various quantities of (+)-catechin solution (2 mg/mL) and manually stirred. Each concentration was then given 1.5 mL of strong hydrochloric HCl and left to react at room temperature for 20 min. The absorbance at 499 nm was measured using a UV-visible spectrophotometer in comparison to a blank. By using the samples instead of the catechin in the calibration curve plotting process, the condensed tannin content of the samples was determined. The calibration curve was utilized to convert the tannin content into milligram equivalents of catechin per gram of dry matter weight.

#### 2.7.5. Determination of Hydrolyzable Tannins

To determine the hydrolyzable tannins, the method of Willis and Allen [47] was used with minor adjustments. In this method, 10 μL of the extract was mixed with 5 mL of 2.5% KIO_3_ and vortexed for ten seconds. The maximal absorbance was attained after 4 min for the standard tannic acid solution and 2 min for the extract and reaction at their best. The absorbance at 550 nm (UV-visible) was measured using a spectrophotometer. The results were represented as mg of tannic acid per gram of dry plant using the calibration curve that was created using 11 different tannic acid concentrations that ranged from 100 to 2000 g/mL.

### 2.8. HPLC/UV ESI-MS Analysis of P. lentiscus Leaves Extracts

High-performance liquid chromatography coupled to Q Exactive Plus mass spectrometry with electrospray as a molecular ionization method (HPLC/UV-ESI-MS) was used to analyze the phenolic compounds of *P. lentiscus* decocted on an UltiMate 3000 HPLC (Thermo Fisher Scientific, Sunnyvale, CA, USA) with a sample changer. The sample changer was set to keep the samples at 5 °C. This HPLC system employed a reverse phase C18 column with a 40 °C column temperature (Lichro CART, Lichrospher, Merck, Darmstadt, Germany, 250 × 4 mm, id 5 μm). The solvents used in the mobile phase were Solvent A: 0.1% formic acid in water (*v*/*v*) and Solvent B: 0.1% formic acid in acetonitrile (*v*/*v*), and the gas was removed using ultrasound. At 20, 25, 26, and 30 min, the gradient’s composition changed from 2% B to 30%, 95%, 2%, and 2% B, respectively. The flow rate was 1 mL/min, and the injection volume was 20 μL. Broadband collision-induced dissociation (bbCID) detection was carried out on a Maxis Impact HD (Bruker Daltonik, Bremen, Germany) after negative electrospray ionization. An L-2455 diode array detector (Merck-Hitachi, Darmstadt, Germany) was also used for UV detection, with scanning in the 190–600 nm range and three acquisition wavelengths of 280–320–360 nm. The capillary voltage of 3000 V, drying gas temperature of 200 °C, dry gas flow rate of 8 L/min, nebulizing gas pressure of 2 bar, and offset plate of 500 V were employed as the parameters. Nitrogen was employed as both the nebulizer gas and the desolvation gas. The MS data’s *m*/*z* range was 100 to 1500. Data collection and analysis were performed using Thermo Scientific’s Chromeleon 7.2 Chromatography Data System (CDS). By looking at the eluted molecules’ mass spectra, the compounds that were eluted were investigated.

### 2.9. Antioxidant Activities

#### 2.9.1. Antiradical Activity by the DPPH^•^ Test

The ability of an antioxidant (phenolic molecule) to donate one electron to the artificial radical DPPH^•^ and stabilize it into DPPH with yellow coloring was used to test the antiradical activity [48]. The experiment was conducted in a UV-visible spectrophotometer at a wavelength of 515 nm. We generated a 6 × 10^−5^ M DPPH^•^ solution by combining 2.4 mg of DPPH with 100 mL of ethanol. The extract samples were prepared by dissolving them in pure ethanol. Before the test, 2.8 mL of the DPPH^•^ solution was combined with 200 L of an extract (sample) or a common antioxidant (ascorbic acid). After 30 min of incubation at room temperature in the dark, the absorbance at 515 nm was assessed in comparison to a blank that contained only ethanol. The data were then translated into percent inhibition using Equation (2) below and the DPPH^•^ solution without extract as the negative control [49].
(2)$$\%AA=\frac{Abscontrol-Abssample}{Abscontrol}\times 100$$
where: Abs sample: absorbance of the test compound (extracts); Abs control: absorbance of the blank (optical density of the solution consisting of DPPH^•^ and ethanol); % AA: percentage of antiradical activity

#### 2.9.2. Ferric Reducing Antioxidant Power (FRAP) Method

According to Oyaizu’s method from 1986, the iron-reducing activity of our extracts was evaluated by measuring the conversion of Fe^3+^ in the K_3_Fe(CN)_6_ complex to Fe^2+^. Samples from the various extracts were examined to do the assessments of antioxidant activity by the FRAP method, following the steps outlined below. The identical procedures as for the samples mentioned above were applied to a certain amount of extract. The absorbance value was measured at 700 nm against a prepared blank (UV-Vis spectrophotometer) after the instrument was calibrated with distilled water. Ascorbic acid, whose absorbance was measured under identical circumstances as the samples, was employed as the positive control in place of a conventional antioxidant solution. The investigated extracts’ lowering power increased along with increases in absorbance [50,51].

#### 2.9.3. Total Antioxidant Capacity (TAC)

The phosphomolybdenum procedure outlined by Khiya [52] was used to calculate the TAC of the extracts. Molybdenum Mo (VI), which is present in this procedure as molybdate ions MoO_4_^2−^, is reduced to molybdenum Mo (V) MoO^2+^ when the extract is present. This molybdenum Mo (V) complex subsequently develops at an acidic pH and is known as a green phosphate/Mo (V) complex. An aliquot of 0.3 mL of the reagent solution (0.6 M sulfuric acid, 28 mM sodium phosphate, and 4 mM ammonium molybdate) was added to each extract along with 3 mL of the reagent solution. After fastening the tubes, 95 °C was applied for 90 min. The 695 nm absorbance of the solutions was measured after cooling, and the results were compared to a generated blank that had been incubated under the same conditions as the sample. As a reference range, several ascorbic acid concentrations were developed. The TAC is reported as milligram equivalents of ascorbic acid per gram of crude extract (mg EAA/g).

### 2.10. Determination of the Minimum Inhibitory Concentration, Minimum Bactericidal Concentration, and Minimum Fungicidal Concentration

The reference microdilution method was used in 96-well microplates to determine the minimum inhibitory concentration (MIC) [53]. The MIC is the lowest concentration of EO required to totally halt the growth of the tested microorganism, as determined by the quantity of growth that was apparent to the naked eye during incubation. To reach concentrations of 5 to 0.93 × 10^−2^ mg/mL for each EO, a series of dilutions were performed using a stock solution of the EO prepared in 10% DMSO. These dilutions were created with a final volume of 100 μL for each concentration in Mueller–Hinton medium for bacteria and Sabouraud broth for fungi. Then, 100 μL of microbial inoculum was added to the various dilution stages with a final concentration of 10^6^ or 10^4^ CFU/mL for bacteria or fungus, respectively. After a 24 h incubation at 37 °C, ten microliters of resazurin were added to each well to gauge bacterial growth. After a second incubation at 37 °C for two hours, the hue changed from purple to pink, signifying microbial development. The lowest concentration that prevents resazurin from changing color was found to be the MIC value. Wells 11 and 12 served as the growth and sterility controls, respectively. The test was run twice on this oil. The standard antifungal used in the study, terbinafine 250 mg, was powdered and combined with 2 mL of 10% DMSO. The minimum bactericidal concentration (MBC) and minimum fungicidal concentration (MFC) were determined by extracting 10 μL from each well that showed no evidence of growth and plating it for 24 h at 37 °C on Mueller–Hinton (MH) agar for bacteria or in Sabouraud broth for fungi. The MBC and MFC are the sample concentrations at which a 99.99% reduction in CFU/mL compared to the control was achieved. The MBC/MIC or MFC/MIC ratio can be used to determine each extract’s effectiveness against microorganisms. As a result, if the ratio is less than 4, the EO has bactericidal/fungicidal activity, and if it is greater than 4, it has a bacteriostatic/fungistatic effect [54].

### 2.11. Anticoagulant Activity

Chronometric coagulation as-says using the prothrombin time and partial thromboplastin time were employed to gauge the anticoagulant effect according to the method described by Hmidani et al. [55]. The prepared decocts were examined in case anticoagulant agents may be investigated; 11.500, 5.750, 2.875, 1.438, 0.719, 0.359, and 0.179 mg/mL of extract were utilized in the coagulation mixtures. Blood was drawn into tubes containing 3.8% trisodium citrate in a polypropylene container. They were immediately separated and centrifuged at 25,000 rpm for 10 min to produce a plasma pool. The recently created plasma pool was maintained at 10 °C before usage. Each sample’s partial thromboplastin time (aPTT) was determined by combining a plant extract solution (50 μL) with 50 μL of citrated normal plasma pool and incubating the combination for 10 min at 37 °C. The PTT reagent (CKPREST^®^), a gift from Diagnostica Stago (Diagnostica Stago, France), was then added to the mixture and incubated for 5 min at 37 °C. Coagulation was induced by adding 25 mmol/L CaCl_2_ (100 μL), and the coagulation time was recorded. Contrarily, in the prothrombin time (PT) experiment, 50 μL of citrated normal plasma pool and 50 μL of a plant extract solution were combined, and the mixture was then incubated for 10 min. Neoplastin^®^ Cl reagent in 200 μL was preincubated at 37 °C for 10 min after the addition. Then, the clotting time was recorded. The various aqueous plant extracts’ anticoagulant properties were swiftly assessed at a range of concentrations. Six measurements were made automatically using a coagulometer (MC4Plus MERLIN Medical^®^, Germany).

### 2.12. Antidiabetic Activity

#### 2.12.1. Study of the Inhibitory Effect of Aqueous Extracts on the Activity of Pancreatic α-Amylase, In Vitro

The concentrations of acarbose that were examined were 1, 0.8, 0.6, 0.4, and 0.2 mg/mL. At doses of 0.89, 0.45, 0.22, 0.11, and 0.06 mg/mL, the aqueous extract was evaluated. The Daoudi et al. [56] method was employed to examine the aqueous extract’s potential inhibitory impact on the enzymatic activity of α-amylase. The phosphate buffer solution was mixed with either 200 μL of the acarbose solution (positive control) or 200 μL of the aqueous extract solution. All tubes received an addition of the enzyme solution except the blank tube, which received 200 μL of phosphate buffer instead. The tubes underwent a 10-min preincubation at 37 °C. The starch solution was then added to each tube in a volume of 200 μL. The entire set was incubated for 15 min at 37 °C. The tubes were filled with 600 μL of DNSA to halt the enzymatic activity. The tubes were then submerged for 8 min in a saucepan of boiling water. Then, thermal shock was used to interrupt this process. Before receiving 1 mL of diluted water, each tube was submerged in an ice water bath. Using a spectrophotometer and a blank made up of the buffer solution rather than the enzyme solution, the absorbance was measured at 540 nm. Using Equation (3) below, the (%) inhibition of each extract or acarbose was determined.
(3)$$\%\;Inhibition=\frac{Acontrol-Asample}{Acontrol}\times 100$$
where: A sample: absorbance of enzymatic activity in the presence of extract or acarbose; A control: absorbance of enzyme activity without inhibitor.

#### 2.12.2. Study of the Inhibitory Effect of Aqueous Extracts on the Activity of α-Glucosidase, In Vitro

The procedure described by Chatsumpun et al. [57] was slightly altered to assess the extract’s inhibitory effect against α-glucosidase using pNPG substrate. The concentration range used to test the extracts was 0.488 to 100 μg/mL. Each sample was prepared using DMSO (5%) and the α-glucosidase enzyme (pH 6.8) with phosphate buffer. Acarbose served as the positive control, and DMSO (5%) was used as a solvent control. A 96-well plate was filled with 10 μL of each sample and 40 μL of the enzyme α-glucosidase (0.1 U/mL). Then, this combination was preincubated at 37 °C for 10 min. The mixture was then incubated at 37 °C for 20 min with 50 μL of pNPG (1 mM) added. Following this, 100 μL of Na_2_CO_3_ (0.1 M) solution was added to stop the reaction. The mixture’s absorbance was measured at a wavelength of 405 nm using a microplate reader. To calculate the proportion of -glucosidase inhibition, Equation (3) was employed.

#### 2.12.3. Acute Toxicity Study

The purpose of this study was to show that the therapeutic dose has no short-term harm in normal mice. We looked at oral toxicity because it is typically involved in acute toxicity in humans under everyday circumstances. The Organization for Economic Cooperation and Development’s (OECD) regulations were followed in conducting this study. *P. lentiscus* extract (E_0_), which was chosen for the pharmacological investigation, was examined at doses of 0.5, 1, and 2 g/kg. Albino mice (20–35 g, 14 h fasted) from two batches were arbitrarily split into four groups (n = 6; ♂/♀ = 1): Aqueous extract E_0_ (0.5 g/kg), Aqueous extract E_0_ (1 g/kg), and Aqueous extract E_0_ (2 g/kg) are employed, with distilled water (10 mL/kg) serving as the control. Before the test started, the mice were weighed. They were then immediately administered one dose of the aqueous extract. Following that, we kept an eye on them for a total of 10 h to search for any signs of obvious poisoning. For the next 14 days, the mice were monitored every day to look for any brand-new clinical or behavioral signs of injury.

#### 2.12.4. Study of the Antihyperglycemic Activity of the Aqueous Extract of *P. lentiscus* in Normal Rats In Vivo

The various test materials were given to normoglycemic mice to conduct the ex vivo oral glucose tolerance test (OGTT) or oral sucrose tolerance test (OSTT). The purpose of this investigation was to determine if normal rats that had been overfed with D-glucose experienced any postprandial antihyperglycemic effects from the extract (E_0_). After a 14 h fast, the normal rats (200–250 g) were divided into three groups (n = 6; ♂/♀ = 1) and given the following treatments: a control dose of distilled water (10 mL/kg), glibenclamide (2 mg/kg), and aqueous extract E_0_ (2 mL/kg). The test substance (distilled water, aqueous extract, or glibenclamide) was then given orally to the normal rats after they had been given ether (inhalation) anesthesia first. This was followed by the collection of a blood sample from the queue to assess blood glucose at time zero. A second blood glucose reading was taken 30 min later, and the rats were then given a D-glucose (2 mg/kg) excess right away. Following that, the variation in blood glucose was observed for 3 h at 30, 60, 90, and 150 min.

### 2.13. PASS, ADMET, and Prediction of the Efficacy of Potentially Active Compounds Isolated from the EO and Aqueous Extract of P. lentiscus Leaves

The main compounds of the EO (spathulenol; germacrene D; bicyclogermacrene; terpinen-4-ol; α-cadinol and globulol) and aqueous extract E_0_ (3,5-di-*O*-galloyl quinic acid; 3-galloyl quinic acid; gallic acid; 3,4,5-tri-*O*-galloyl quinic acid; gallic acid 3-*O*-gallate; myricetin-*O*-xyloside and trigallic acid) of *P. lentiscus* were selected for PASS and ADMET (Absorption, Distribution, Metabolism, Excretion, and Toxicity) prediction studies. ChemBioDraw (PerkinElmer Informatics, Waltham, MA, USA, v13.0) [58] was used to select the SMILES format for these compounds. Simulations were then performed using the SwissADME [59] and pkCSM [60] web tools for ADMET prediction, as well as the PASS-Way2Drug web prediction tool [61]. PASS denotes “drug-like” compounds’ potential activity (Pa) and likely inactivity (Pi) [62]. To investigate toxicity levels and obtain relevant data on various toxicological parameters such as LD_50_ and toxicity class, we used Pro-tox II (https://tox-new.charite.de/protox_II/, accessed on 10 August 2023), a valuable tool specifically designed for this purpose [63]. Physicochemical characteristics, lipophilicity, water solubility, pharmacokinetics, drug-likeness, medicinal chemistry, and toxicological properties for the selected chemicals (ligands) were predicted using the ADMET program (SwissADME, pkCSM, and Protox II), which was used to evaluate the selected substances (ligands). By employing these analytical methodologies and tools, we have gained compelling findings regarding the potential therapeutic applications and potential side effects associated with the major chemical components identified in *P. lentiscus* EO and decoction.

### 2.14. Molecular Docking Prediction

The binding capacity of a ligand to the target receptor’s binding site for various conformations and positions of the ligand can be assessed using the crucial computational tool known as molecular docking [64]. The RCSB Protein Data Bank (www.rcsb.org, accessed on 15 May 2023) was used to obtain the three-dimensional structures of the proteins NADPH oxidase (PDB ID: 2CDU), thrombin (PDB ID: 4UFD), α-amylase (PDB ID: 4W93), and α-glucosidase (PDB ID: 3W37). The PubChem database (www.pubchem.ncbi.nlm.nih.gov, accessed on 15 May 2023) provided the ligands’ crystal structures in sdf format, which were subsequently translated to PDB format using the online Babel program [65]. Using the proper force field, these ligands were reduced and energetically optimized [66,67]. Water was taken out of the receptor structures, and then polar hydrogen atoms and Kollman charges were added. To determine the binding affinities of proteins and ligands, AutoDockTools-1.5.6 (ADT) simulations were run [68]. Several input files, including ligands and proteins and pdbqt ligand and proteins, were produced in order to execute ADT. AutoDock 4.2 was used to calculate protein-ligand docking. Using PyMOL 2.5.2 software, the RMSD of heavy atoms between docked poses and crystallographic poses of ligands was determined [69] to assess the accuracy of docking. If the root-mean-square deviation (RMSD) is less than 2 Å, then a molecular docking approach is considered credible [70]. Then, using the Discover Studio 2021 viewer (v21.1.0), the optimal poses were chosen based on docking scores, and interactions between residue and ligand were looked at and depicted [71].

### 2.15. Statistical Analysis

The mean and standard error of the mean are used to express the results. The one-way ANOVA test of variance and Tukey’s post-test were used to evaluate the results using GraphPad Prism 9 software (version 9.5.1, San Diego, CA, USA). *p* values lower than 0.05 were regarded as significant. Using R software (version 4.1.3), the relationship between phenolic component concentrations and antioxidant activity was examined.

## 3. Results

### 3.1. Quality Control of Plant Material

The results of the quality control of plant material are presented in Table 4. The *P. lentiscus* leaves have a moisture content of about 10.12%, which is much less than the 12% upper limit. The pH of our plant is acidic (3.52; acidophile), and it contains 5.47% mineral matter. Thus, heavy metal analysis in the genus *Pistacia* has received very little attention. In our case, a total of 7 elements (Cr, Cd, As, Fe, Pb, Ti, and Sb) were determined. The *P. lentiscus* sample revealed levels that were below the FAO/WHO-required limits.

### 3.2. Yield of P. lentiscus EO

The average essential oil yield was calculated based on the dry plant material obtained from *P. lentiscus* leaves. The EO yield obtained was very low, around 0.22 ± 0.01% (Table 5). This oil is characterized by a transparent color with an aromatic odor.

### 3.3. GC-MS Analysis of P. lentiscus EO

GC-MS analysis of the EO from *P. lentiscus* leaves allowed for the determination of the chromatographic profile (Figure 2), identification of the different constituents, and measurement of their relative abundances in the analyzed essential oil (Table 6).

The identification of the chemical composition of *P. lentiscus* EO enabled us to identify a total of 33 compounds, representing 99.95% of the total chemical composition (Table 6).

We observed that the leaves of the pistachio are mainly composed of oxygenated sesquiterpenes (45.07%), hydrocarbon sesquiterpenes (35.87%), and oxygenated monoterpenes (15.17%). The major compounds are spathulenol (18.57%), germacrene D (17.54%), bicyclogermacrene (12.52%), and terpinen-4-ol (9.95%). 

Based on the results presented in Figure 3, we observe that the EO from *P. lentiscus* leaves is rich in non-aromatic alcohols (52.83%) and hydrocarbons (39.05%), followed by esters (4.14%), fatty acids (3.40%), and traces of epoxides (0.53%).

### 3.4. Phytochemistry Screening

*P. lentiscus* leaves have been shown to have significant secondary metabolites according to phytochemical analyses (Table 7). A very positive reaction was seen, pointing to the existence of sterols and triterpenes, flavonoids, anthocyanins, tannins, mixed anthracene derivatives, saponins, sugars and holosides, and mucilages. Alkaloids and quinones also caused a negative response.

### 3.5. Extraction and Quantitative Analysis of Phenolic Compounds

#### 3.5.1. Extraction Yields

To compare the yields and contents of phenolic compounds produced by these two procedures, we carried out solid-liquid extractions by decoction (E_0_), Soxhlet water (E_1_), and Soxhlet ethanol-water (E_2_). According to Figure 4, extracts from *P. lentiscus* leaves obtained through decoction appear to have a higher yield than those produced through Soxhlet extraction.

#### 3.5.2. Determination of Phenolic Compounds

To evaluate the contents of polyphenols, flavonoids, condensed tannins, and hydrolyzable tannins in aqueous and organic extracts of *P. lentiscus*, we established calibration curves for gallic acid (Y = 0.095X + 0.003; R^2^ = 0.998), quercetin (Y = 0.073X − 0.081; R^2^ = 0.995), catechin (Y = 0.7421X + 0.0318; R^2^ = 0.998), and tannic acid (Y = 0.1700X − 0.0006718; R^2^ = 0.996). According to milligram equivalents of gallic acid, quercetin, vanillin, and tannic acid per gram of extract, the levels of total polyphenols, flavonoids, condensed tannins, and hydrolyzable tannins in the extracts were each assessed.

Total polyphenol quantification results demonstrate that the amounts of these compounds differ significantly from one extract to the next (Figure 5). The Soxhlet aqueous extract of *P. lentiscus* leaves had the highest concentration of polyphenols (934.737 mg EAG/g), followed by the decocted and then the hydroethanolic extract. The flavonoid content values (Figure 5) demonstrate the significant flavonoid richness of *P. lentiscus* extracts, with Soxhlet’s aqueous extract recording the highest levels (59.905 mgEQ/g), followed by the hydroethanolic extract (17.521 mgEQ/g), and finally the decoction (13.053 mgEQ/g). *P. lentiscus* leaves are extremely high in tannins, particularly hydrolyzable tannins, according to an examination of the tannin findings. The decocted had a lower concentration of condensed and hydrolyzable tannins than the extracts that Soxhlet obtained from *P. lentiscus*.

#### 3.5.3. Analysis and Identification of Polyphenols in *P. lentiscus* Extracts by High-Pressure Liquid Chromatography-Mass Spectrometry (HPLC/UV-ESI-MS)

The various chemicals found in *P. lentiscus* leaves are shown in the chromatogram in Figure 6 from the HPLC/UV-ESI-MS analysis of the decocted extract (E_0_). In addition to the chromatogram, the examination of the mass spectra allowed us to propose 31 compounds, which are shown in Table 8.

The analytical study of the mass spectra of the aqueous extract of *P. lentiscus* in negative mode shows the presence of several phenolic acids, polyphenols, and flavonoids. The main phenolic acids identified in the decocted are 3,5-Di-*O*-galloyl quinic acid (26.41%), 3-Galloyl quinic acid (18.01%), Gallic acid (13.61%), 3,4,5-Tri-*O*-galloyl quinic acid (11.13%), Gallic acid 3-*O*-gallate (8.23%), Trigallic acid (3.84%), and Caffeic acid 3-glucoside (2.76%). The chemical structures of the main compounds identified in *P. lentiscus* leaves are illustrated in Figure 7.

The use of electrospray mass spectrometry (ESI-MS) detection in negative ion mode allows for mass fragmentation of each compound as an additional method to confirm their structure. Regarding the fragmentation of the identified major compounds with a high percentage, we can mention the following:3,5-Di-*O*-galloyl quinic acid: was identified by its pseudo-molecular peak at *m*/*z* = 495 [M-H]^−^ in negative mode. Scanning of the ion produced by the deprotonated molecule [M-H] showed the typical loss of a galloyl unit (152 Da), resulting in an ion at *m*/*z* 343 [M-H-152]^−^ as a characteristic fragment of 3-Galloyl quinic acid, followed by the loss of a second galloyl molecule, resulting in the ion at *m*/*z* 191 [M-H-304]^−^ as a characteristic fragment of quinic acid.Gallic acid: was identified at 7.69 min by its pseudo-molecular peak at *m*/*z* = 169 [M-H]^−^ in negative mode. Scanning of the ion produced by the deprotonated molecule [M-H] showed the typical loss of CO_2_, resulting in an ion at *m*/*z* 125 [M-H-44]^−^ as a characteristic fragment of pyrogallic acid.3,4,5-Tri-*O*-galloyl quinic acid: also known as trigalloylquinic acid or TGA, is a chemical compound derived from gallic acid and quinic acid. This compound exhibited the quasi-molecular ion [M-H]^−^ at *m*/*z* = 647. It was identified by the production of fragments at *m*/*z* 495, 343, and 169, corresponding to successive losses of gallic acid ([M-152-H]^−^ and [M-304-H]^−^) and the gallic acid fragment, respectively.Gallic acid 3-*O*-gallate: was identified by its pseudo-molecular peak at *m*/*z* = 321 [M-H]^−^ in negative mode, and its peak at *m*/*z* = 169 [M-162]^−^ corresponds to gallic acid after the loss of a galloyl unit (152 Da).

The other ions at *m*/*z* = 449 [M-H]^−^ correspond to the pseudo-molecular ion of myricetin-3-*O*-xyloside, and *m*/*z* = 473 [M-H]^−^ to the quasi-molecular ion of trigallic acid. Thus, the pseudo-molecular ions at *m*/*z* = 341 [M-H]^−^; *m*/*z* = 479 [M-H]^−^; *m*/*z* = 435 [M-H]^−^; *m*/*z* = 609 [M-H]^−^; and *m*/*z* = 799 [M-H]^−^ correspond to the molecules caffeic acid 3-glucoside; myricetin-3-*O*-glucoside; phloretin-2-*O*-glucoside; rutin; and 1,3,4,5-tetra-*O*-galloylquinic acid, respectively.

### 3.6. Antioxidant Activity

The antioxidant activities of the aqueous extract and hydroethanolic extract of *P. lentiscus*, as well as the standard (ascorbic acid), were evaluated using three methods (DPPH, FRAP, and CAT). Calibration curves for ascorbic acid were established for the DPPH method (Y = 1.013X − 8.032; R^2^ = 0.9893), the FRAP method (Y = 0.004760X + 0.09740; R^2^ = 0.8963), and the CAT method (Y = 0.04066X + 0.02110; R^2^ = 0.9949).

The extracts are qualified as natural antioxidants due to their strong capacity to reduce and/or prevent the formation of free radicals. The results from Figure 8A demonstrate that the aqueous extract and hydroethanolic extract of *P. lentiscus* possess a strong anti-radical power. The hydroethanolic extract E (2), the decocted E (0), and the aqueous extract obtained by Soxhlet (E_1_) exhibited very high antioxidant activity with EC_50_ values of 2.630, 7.489, and 12.588 μg/mL, respectively. The EC_50_ of ascorbic acid was 19.378 μg/mL. The results obtained with the FRAP method show that the *P. lentiscus* extracts are characterized by very pronounced EC_50_ values for iron reduction. Figure 8B demonstrates significant differences between the reducing power of the *P. lentiscus* leaf extracts and the positive control (ascorbic acid). The organic extracts obtained by Soxhlet from *P. lentiscus* showed higher reduction capacity compared to the decocted (17.653 μg/mL). However, these results are lower than that of the standard (ascorbic acid) with a concentration of 0.470 μg/mL. Because total antioxidant activity (TAC) is reported in terms of ascorbic acid equivalents, the phosphomolybdenum technique is quantitative. The studied extracts had very strong overall antioxidant capacity, as shown in Figure 8C. The findings demonstrated that organic extracts exhibited higher TAC than decocted compounds, with the aqueous extract recording the most activity (532.953 mgEAA/g), followed by the hydroethanolic extract (425.69 mgEAA/g). Additionally, the findings shown in Figure 9 show an exact correlation between the concentrations of phenolic compounds in the examined extracts and the numerous antioxidant tests that were performed, particularly between the concentrations of polyphenols, flavonoids, and hydrolyzable tannins in relation to the DPPH, FRAP, and TAC tests.

### 3.7. Antimicrobial Activity

Table 9 displays the findings of the antimicrobial activity of *P. lentiscus* decocted and EO. Using the criteria proposed by Sartoratto, Duarte, Wang, Oliveira, and their colleagues, the MIC of the decocted were classed [72,73,74,75]. The level of antimicrobial activity was divided into three categories: high (MIC 0.6 mg/mL), moderate (MIC 0.6–2.5 g/mL), and low (MIC > 2.5 g/mL). The decocted has a higher bactericidal quality than the EO against the investigated species, according to *P. lentiscus* MIC studies. The *P. lentiscus* EO and decocted samples’ MIC, MBC, and MFC studies reveal that the decoction has a high level of bactericidal activity against yeast and enterobacterial species. However, some of the tested strains showed some resistance. The evaluated decocted is powerful against Gram-negative bacilli, particularly against *K. pneumoniae,* with an MIC of 0.6 mg/mL. Furthermore, it is effective against *A. baumannii*, *E. aerogenes*, *E. cloacae*, *P. fluorescence*, *Salmonella* sp., *Shigella* sp., and *Y. enterolitica* with an MIC of 1.2 mg/mL. Furthermore, this decoction is also active against Gram-positive bacilli, particularly against *S. aureus* BLACT, *S. group* D, *S. agalactiae* (B), and *E. faecium* with an MIC of 1.2 mg/mL. Additionally, *P*. *lentiscus* decocted is more active against candida strains, particularly against *C. krusei* and *C*. *albicans,* with an MIC of 0.3 and 2.5 mg/mL, respectively.

### 3.8. Anticoagulant Activity

According to Figure 10, it appears that the studied decoction showed remarkable anticoagulant activity by inhibiting both the endogenous and exogenous coagulation pathways in a dose-dependent manner. In fact, *P. lentiscus* decoction is capable of significantly prolonging the prothrombin time and partial thromboplastin time (*p* < 0.001) at concentrations of 2.875 and 5.750 mg/mL, respectively.

Normal smears and those with the addition of *P. lentiscus* decocted at different concentrations were performed (Figure 11). The results of the hemogram of the samples treated with the decoction show a very low, non-significant reduction in red blood cell count, hemoglobin, and hematocrit compared to the control sample (Table 10). Additionally, this decoction did not affect the platelets since their count, mean platelet volume (MPV), and platelet distribution index (PDI) were stable.

### 3.9. Antidiabetic Activity

#### 3.9.1. Evaluation of the Inhibitory Effect of Decocted Extract on the Activity of α-Amylase and α-Glucosidase, In Vitro

The results shown in Figure 12 for the in vitro testing of *P. lentiscus* decocted inhibition of α-amylase and α-glucosidase demonstrate that the extract produces a varied and dose-dependent inhibitory impact. Acarbose and *P. lentiscus* decocted both exhibited substantial inhibitory effects against α-amylase and α-glucosidase. With an EC_50_ of 364.446 and 17.269 g/mL, respectively, acarbose’s action on α-amylase and α-glucosidase increased in a concentration-dependent manner. *P. lentiscus* decocted had a substantially greater impact on α-amylase and α-glucosidase than acarbose (95.217 mg/mL and 14.643 g/mL, respectively).

#### 3.9.2. Acute Toxicity Study of *P. lentiscus* Decocted Extract

This acute toxicity test’s outcome demonstrates that the decocted is not harmful, even at 2 g/kg. Over the course of the monitoring period, the extract did not result in any deaths or toxic symptoms (such as diarrhea, vomiting, altered movement, etc.).

#### 3.9.3. Study of the Antihyperglycemic Activity of *P. lentiscus* Decocted Extract in Normal Rats In Vivo

Figure 13 and Figure 14 compare the total areas under the blood glucose curve over the course of 150 min and analyze the glucose tolerance test results.

Oral glucose tolerance test

A high peak in blood glucose was visible in normal rats 30 min following a glucose load. Rats given the decocted and glibenclamide showed an improvement in their responsiveness to a glucose load. In comparison to the group of rats pre-treated with distilled water, oral administration of *P. lentiscus* extract at a dose of 400 mg/kg 30 min before to glucose overload significantly reduced postprandial hyperglycemia for this examined extract (Figure 13). In contrast to the group of rats pre-treated with distilled water, glibenclamide significantly reduced postprandial hyperglycemia during the first hour (60 min) after glucose overload (*p <* 0.001; 1.08 g/L).

Areas under the curve (AUC) of postprandial glucose levels.

Rats given decocted extract had a significantly lower area under the curve (*p* ≥ 0.01; 56.93 g/L/h than rats given distilled water (62.91 g/L/h). In addition, animals fed with distilled water had an area under the curve that was significantly (*p* ≥ 0.01) lower than that of glibenclamide (55.95 g/L/h) (Figure 14).

### 3.10. PASS, ADMET, and Prediction of the Efficacy of Potentially Active Compounds Isolated from the EO and Aqueous Extract of P. lentiscus Leaves

In addition to the biological properties, the study of the physicochemical properties of candidate compounds is crucial for the development of therapeutic agents and to justify the effectiveness of the chemical composition of *P. lentiscus* for use as a nutraceutical preservative agent. Thus, the compounds identified in the EO and decocted of *P. lentiscus* leaves were examined to predict their pharmacokinetic and physicochemical parameters and their similarities to drugs.

The PASS prediction of the main compounds isolated from the EO and decocted of *P. lentiscus* leaves was performed to predict the effectiveness of antioxidant, antimicrobial, anticoagulant, and anti-diabetic activities. Table 11 presents the results of the PASS and ADMET prediction studies.

All compounds showed significant values of ‘Pa’ for various activities such as antioxidant (0.151–0.944), antifungal (0.398–0.711), antibacterial potential (0.199–0.585), coagulation effect (0.105–0.253), antithrombotic effect (0.224–0.615), hemostatic effect (0.169–0.939), inhibition of alpha-amylase (0.076–0.776) and inhibition of alpha-glucosidase (0.148–0.522) (Table 11).

According to our predictions, all significant compounds have good antioxidant, hemostatic, and antifungal properties, as well as an inhibitory effect on α-amylase. Additionally, these compounds also have good antithrombotic and antibacterial properties and an inhibitory effect on α-glucosidase but exhibit a low coagulation effect.

The SwissADME and pkCSM web tools are valuable for understanding the physicochemical and pharmacokinetic properties and drug-likeness of compounds. The radar chart of bioavailability, which represents the pharmacokinetic, physicochemical, and drug-likeness properties of compounds present in the EO and aqueous extract (E_0_) of *P. lentiscus*, was analyzed using the Swiss-ADME web server; the results are presented in Figure 15A.

The BOILED-Egg prediction results “Figure 15B” of this study showed that Spathulenol, α-Cadinol, Globulol, Terpinen-4-ol, as well as Gallic acid, have a strong ability to be absorbed by the gastrointestinal tract and passively diffuse through the blood-brain barrier (BBB).

The drug-likeness properties of the selected compounds can be evaluated through the topological polar surface area (TPSA) and molar refractivity. Prediction results showed that significant compounds of the EO as well as Gallic acid from the aqueous extract (E_0_) of *P. lentiscus*, with bioactivity scores of 0.55 or 0.56, all met drug-likeness rules without any violations. The molar refractivity of the majority of studied compounds, except for 3,4,5-Tri-*O*-galloylquinic acid, was within the normal range (40–130). Thus, the synthetic accessibility of these compounds (1.22–5.09) showed a clear synthetic pathway. The lipophilicity values of all selected compounds showed that they were highly soluble in water.

The selected compounds have powerful Caco-2 permeability values and good skin permeability values (log Kp). Thus, most compounds have very good intestinal absorption (HIA > 30%) except for 3-Galloylquinic acid, 3-5-Di-*O*-galloylquinic acid, and 3,4,5-Tri-*O*-galloylquinic acid. P-glycoprotein (P-gp) plays an important role in the absorption and distribution of drugs. The major compounds of the EO, as well as Gallic acid (decocted), were not found to be P-gp substrates, while the other compounds in the decocted will have increased bioavailability. Thus, all studied compounds are not inhibitors of P-gp I and P-gp-II.

The isolated molecules from the EO with log BB > 0.3 and CNS score > −3.0 have the ability to easily penetrate the blood-brain barrier (BBB) and weakly penetrate the CNS, with variable volumes of distribution (logVDss) ranging from 0.210 L/kg to 0.648 L/kg, in tissues. On the other hand, the isolated molecules from the decocted (log BB < −1 and CNS score < −3.50) are poorly distributed in the brain and do not penetrate the CNS. Cytochrome P450 (CYP) enzymes and molecular interactions are essential for drug elimination; Globulol and Bicyclogermacrene inhibit CYP1A2. Thus, Spathulenol inhibits CYP2C19 iso-enzymes, leading to adverse effects and drug interactions. Therefore, adverse effects resulting from drug interactions during oral administration of other compounds are unlikely.

To predict the excretion pathway, total clearance (CLTOT) for hepatic and renal substrates and renal organic cation transporters 2 (OCT2) were expressed as predicted log mL/min/kg. The findings showed that most phytochemical components had a positive overall clearance value and may be eliminated. 

The toxicity profile of the phytochemical elements discovered in *P. lentiscus* EO and decoction was projected using the toxicity criteria of AMES, hERG potassium channel inhibition, skin sensitization, hepatotoxicity, carcinogenicity, immunotoxicity, mutagenicity, and cytotoxicity.

Compounds with an LD_50_ between 2000 and 5000 mg/kg are considered potentially harmful if ingested (GHS hazard class V), while those with an LD_50_ between 300 and 2000 mg/kg are classified as harmful if ingested (GHS hazard class IV). Based on the LD_50_ values provided in Table 11, the majority of compounds have a moderate level of acute toxicity and fall into GHS hazard classes IV and V. However, none of the compounds are classified as highly toxic (GHS hazard classes I-III) since all have LD_50_ values greater than 300 mg/kg.

According to the results, none of the chemicals derived from *P. lentiscus* showed the potential to induce hepatotoxicity, mutagenicity, or cytotoxicity, indicating that they were all relatively safe to use, except for the compounds isolated from the essential oil, which may cause skin sensitization effects. Gallic acid was found to be carcinogenic in terms of its ability to induce cancer, but its probability of occurrence was less than 0.56%, indicating a negligible propensity to cause cancer. On the other hand, the immunotoxicity of α-cadinol (0.69%), germacrene D (0.80%), myricetin-3-*O*-xyloside (0.96%), and 3-galloylquinic acid (0.63%) was predicted with a probability demonstrating the potential of these substances, particularly germacrene D and myricetin-3-*O*-xyloside, to cause immunotoxicity.

### 3.11. Molecular Docking

In this study, since the EO and aqueous extract of *P. lentiscus* showed substantially higher in vitro bioactivities, the compounds identified by GC-MS and HPLC/UV-ESI-MS were selected for in silico molecular docking studies.

In the molecular docking experiments, the antioxidant, antifungal, antibacterial, anticoagulant, and antidiabetic activities of the compounds and their probable mode of action were deduced on respective target proteins (NADPH-oxidase (2CDU), thrombin (4UFD), α-amylase (4W93), and α-glucosidase (3W37)) via their molecular interaction at the atomic level. NADPH oxidases use the catalytic subunit Nox to catalyze the transfer of electrons from NADPH to molecular O_2_ to generate ROS, such as superoxide or H_2_O_2_. This enzyme is considered important for the development of antioxidants and antibiotics. Thus, it has been reported that obesity promotes a chronic inflammatory and hypercoagulable state that favors cardiovascular diseases, while increased thrombin activity plays a key role in thromboembolic events related to obesity. These results inspired us to study whether the major constituents of *P. lentiscus* can modulate thrombin’s proteolytic activity. Furthermore, natural source α-amylase and α-glucosidase inhibitor compounds have shown potential responses in managing hyperglycemia and have attracted researchers from all over the world.

The molecular docking study specifically focused on free binding energy, hydrogen bonds, carbon–hydrogen (C–H) bonds, and Van der Waals (VDW) interactions. Hydrogen bonds and VDW are associated with binding interactions, while C–H bonds and pi-sigma interactions are associated with the stability of the ligands (selected compounds) and the docked receptor complex. The docking binding energy scores among the binding sites of target proteins (NADPH-oxidase, thrombin, α-amylase, and α-glucosidase) are presented in Table 12. Additionally, the H bonds, C–H bonds, and VDW interactions with the amino acids involved in the binding sites of NADPH-oxidase, thrombin, α-amylase, and α-glucosidase were studied. Furthermore, Table 12 shows the results of ligand evaluation for their binding energy with different target proteins. Trigallic acid showed the strongest binding energy with α-amylase (−8.4) and NADPH-oxidase (−11.2), while 3-5-Di-*O*-galloylquinic acid showed the strongest binding energy with thrombin (−8.1) and α-glucosidase (−8.1), while 3,4,5-Tri-*O*-galloylquinic acid showed the strongest binding energies with α-amylase (−8.4) and α-glucosidase (−8.1), as well as strong binding energy with NADPH-oxidase protein (−10.4).

The docking simulation of the different ligands studied with the crystal structures of NADPH-oxidase, thrombin, α-amylase, and α-glucosidase during in silico molecular interaction is presented in Figure 16 and Table 13.

The interaction of trigallic acid with NADPH-oxidase and α-amylase was related to hydrogen and C–H bonds and mostly associated with 6 VDW interactions with each protein (HIS10, ASP282, THR112, LYS134, GLU32, HIS79, and ASP356, GLY304, ARG303, ASP353, HIS305, 3L9503, respectively) (Figure 16 and Table 13).

The interaction of 3-5-Di-*O*-galloylquinic acid with thrombin and α-glucosidase was related to hydrogen and C–H bonds and mostly associated with 5 VDW interactions with each protein (SER195, LEU41, TRP141, ASN143, ASP60E and ARG699, PHE680, ASP305, ARG670, SO41010, respectively) (Table 13).

However, the docking of 3,4,5-Tri-*O*-galloylquinic acid with digestive enzymes (α-amylase and α-glucosidase) indicated eight VDW interactions (GLN7, ARG10, GLY9, ASP402, THR11, ARG398, ARG252, THR6) and three VDW interactions (THR681, ARG814, ASP305), as well as other non-covalent interactions (Table 13).

## 4. Discussion

The preservative properties of aromatic and medicinal plants are highly sought after due to their traditional use in folk medicine. According to our ethnobotanical survey conducted in the Boulemane region, we found that *P. lentiscus* is a potential source of antioxidant agents [76].

Our research sought to isolate and describe substances from Moroccan *P. lentiscus* EO, aqueous, and organic extracts, as well as assess their antioxidant, antibacterial, anticoagulant, and antidiabetic effects in vitro, in vivo, and in silico.

The values obtained regarding the analysis of physicochemical parameters of *P. lentiscus* leaf powder did not exceed the standards described in the European Pharmacopoeia. This confers better long-term preservation to the studied powders.

The EO yield obtained from *P. lentiscus* leaves in our work remains low but higher than the yields reported by Ailli in Khenifra, Morocco (0.17%) [19], or those recorded by Bachrouch in Tunisia, particularly in Oued El Bir (0.009%), Jebel Mansour (0.02%), Siliana (0.007%), and Tabarka (0.01%) [77].

The primary components of *P. lentiscus* leaf EO have similarities and differences, according to earlier investigations on their chemical makeup. According to Chaabani in 2020 [78], the primary constituents of the EO extracted from *P. lentiscus* leaves in Tunisia were germacrene D (16.58%), δ-cadinene (13.60%), β-caryophyllene (11.06%), bornyl acetate (7.52%), 1-terpinen-4-ol (5.43%), α-cadinol (4.24%), and α-pinene (3.53%). These findings agree with ours, especially given that germacrene D was the dominant substance. However, our findings do not agree with earlier research on Tunisian *P. lentiscus* EO. For instance, Ben Douissa et al. [79] demonstrated that terpinen-4-ol (12%), α-pinene (17%), and γ-terpinene (9%) were the three main components of the EO from Tunisian *P. lentiscus*. Terpinen-4-ol, β-caryophyllene, and α-terpineol were also listed as the primary chemicals in pistachio tree leaves by Bachrouch et al. [80]. These variations in EO composition may be explained by a number of variables, including plant age, developmental stage, location, and harvesting time.

Compared to the decocted extract, the hydroethanolic extract and aqueous extract generated by Soxhlet extraction had much larger variations in phenolic compound content, according to the results of phenolic compound quantification. *P. lentiscus* leaves had a slightly lower concentration of flavonoids and tannins but a greater concentration of total phenolic compounds than those in Algeria reported by Remila [81] and Mehenni [82]. Additionally, this study presents the first comparison of three distinct extracts produced by decoction, Soxhlet extraction, and ethanol/water combination. These findings demonstrate the high total polyphenol, flavonoid, and tannin content of *P. lentiscus* leaves gathered in the Boulemane region, which suggests that they may have important biological and pharmacological effects.

Furthermore, the results of the chromatogram and mass spectrometry analysis (HPLC/UV-ESI-MS) obtained were significantly consistent with previous studies [83], [84,85]. The provided information offers reliable confirmation of the structures based on mass spectral analysis of 3,5-Di-*O*-galloyl quinic acid, 3-Galloyl quinic acid, Gallic acid, 3,4,5-Tri-*O*-galloyl quinic acid, Gallic acid 3-*O*-gallate, Trigallic acid, and Caffeic acid 3-glucoside.

We assessed the antioxidant activities using three methods (DPPH, FRAP, and TAC) in order to assess the antioxidant capacity of *P. lentiscus* extracts. The outcomes of these experiments showed how highly concentrated in antioxidant molecules the *P. lentiscus* extracts under study were. Because *P. lentiscus* extracts contain a lot of phenolic chemicals, they have a powerful anti-free radical effect. In fact, they are more effective than ascorbic acid, a common antioxidant, at scavenging the DPPH radical. The chemical makeup of the extracts—complex combinations of phenolic compounds—can be linked to this intriguing biological function. There is a predominance of phenolic acids among these compounds, particularly those found in *P. lentiscus* decocted by HPLC/UV-ESI-MS, such as 3,5-Di-*O*-galloyl quinic acid, 3-Galloyl quinic acid, Gallic acid, 3,4,5-Tri-*O*-galloyl quinic acid, Gallic acid 3-*O*-gallate, Trigallic acid, and Caffeic acid 3-glucoside.

Our findings outperform those in the literature, such as those from research by Cardullo in Italy [86] and Elloumi in Tunisia [87]. Figure 9 illustrates a considerable positive association between the number of total phenols and the outcomes of antioxidant testing. The majority of the antioxidant activity of *P. lentiscus* extract is attributed to gallic acids and their galloyl derivatives (5-*O*-galloyl, 3,5-*O*-digalloyl, and 3,4,5-tri-*O*-galloyl). With more galloyl groups present, more DPPH is scavenged [88]. Quercetin and gallic acid are examples of polyphenols that are more efficient than monophenols [89,90]. Gallic acid’s ability to boost antioxidant activity is significantly influenced by the inductive impact of hydroxyl groups [89]. This indicates that these substances play a role in the antioxidant capacity found in *P. lentiscus* extracts [81].

The polarity of the solvent, the solubility of phenolic compounds, and the hydrophobicity of the reaction media are additional factors that affect antioxidant activity. Polar chemicals, which serve as hydrogen atom donors or single atom transfer agents, are abundant in polar extracts such as ethanol, methanol, and aqueous extracts [91]. The degree of phenol polymerization, the plant component used, and differences in soil and climatic conditions are other factors that affect phenolic compound solubility in addition to solvent polarity [92,93].

Research on the antibacterial properties of *P. lentiscus* extracts has so far been covered in a number of international studies. These investigations’ findings indicate that *P. lentiscus* leaf extracts show intriguing antibacterial activity against a variety of fungi and bacteria, including Gram-positive and Gram-negative bacteria. Methods using disk diffusion and/or dilution were discovered to be the most effective. The effectiveness of lentisk leaf extracts against microorganisms, including *Micrococcus luteus*, *Bacillus subtilis*, *Listeria innocua*, and *Escherichia coli*, was found to be independent of the solvent used to make them (dichloromethane, methanol, ethanol, ethyl acetate, and water) [91]. The same extracts also demonstrated inhibitory effects against *Candida pelliculosa* and *Fusarium oxysporum* albidini fungal strains [91]. Lauk et al. [94] demonstrated that *P. lentiscus* leaf decocted had good antibacterial activity against *Staphylococcus aureus* and *Escherichia coli* (with MIC = 312 mg/L for both tested bacteria). A more significant antibacterial effect was seen against fungal cells such as *Candida albicans*, *Candida parapsilosis*, *Torulopsis glabrata*, and *Cryptococcus neoformans* [93]. With a MIC value of 0.3 mg/mL, Bakli et al. research [95] demonstrated the outstanding effectiveness of an ethanol leaf extract against *Vibrio cholerae*. Our findings are in line with those of this other research, where it was discovered that Gram-negative bacteria and candidiasis were more sensitive to the investigated *P. lentiscus* extracts than Gram-positive bacteria. Variations in the examined microorganisms’ cell wall structures may be to blame for this discrepancy.

The typical hemostatic procedure stops bleeding from a cut or wound on blood arteries by generating a platelet thrombus; once healing is complete, the thrombus finally dissolves. The interaction of platelets and coagulation factors with the blood arteries is crucial to this intricate multiphase system. A flaw in any of these steps can lead to thrombosis or hemorrhage [96]. In this investigation, the decocted considerably prolonged the prothrombin time (PT) and activated partial thromboplastin time (aPTT), demonstrating a clear distinction between coagulation and anticoagulation.

Research on the anticoagulant and antithrombotic properties of Pistacia species is currently lacking. However, it is well-recognized that naturally occurring phenolic chemicals can affect platelet aggregation and coagulation. The impact of methanol extract from *Angelica shikokiana* on coagulation was examined by Mira et al. [97]. Quercetin, luteolin, and kaempferol, which were studied as isolated polyphenols from this plant, were found to considerably extend the aPTT and PT, indicating inhibition of the intrinsic, extrinsic, and common pathways of the coagulation system. In a different study, rats received an intravenous injection of rosmarinic acid purified from *Radix salviae* five to ten minutes before blood was taken. After rosmarinic acid injection, analysis of the coagulation parameters revealed a prolonging of the aPTT, PT, and recalcification time and suppression of collagen-induced platelet aggregation [98]. Additionally, a 2009 study by Chao examined the effects of caffeine on diabetic rats and found that plasma antithrombin III levels significantly increased, indicating that caffeine has an anticoagulant effect [99]. Less research has been done on the methods by which polyphenols prevent the coagulation cascade. It is evident that these metabolites, which include coagulation factors as serine proteases, impede their function [100]. For the three enzymes thrombin, trypsin, and urokinase, the effects of certain flavonoids on the activity of serine proteases in vivo have been investigated. The three enzymes responded differently to the tested flavonoids. For instance, quercetin, used as a reference, had a CE_50_ value of 30 μM for inhibiting thrombin. Among the 14 studied flavonoids, salicin had the strongest thrombin inhibitory activity, with a CE_50_ of 11.4 μM [101]. Quercetin effectively suppressed the activity of activated factor X by a competitive mechanism, with a CE_50_ value of 5.5 ± 0.6 μM, according to a study that examined the effects of 20 prevalent polyphenols, including rutin, hyperoside, and caffeic acid [102].

When compared to acarbose and other published research, the decocted extract’s antidiabetic effect against digestive enzymes demonstrated stronger inhibitory activities against α-glucosidase and α-amylase [82,103]. The capacity of the aqueous extract from *P. lentiscus* to block the enzymes α-glucosidase and α-amylase supports its historical usage in the treatment of diabetes. This suggests that the compounds with antidiabetic activity are extracted in water. Prior research has demonstrated that phenolic substances have a hypoglycemic effect, increasing postprandial blood glucose levels, acute insulin production, and insulin sensitivity [104]. Previously, it was believed that the only way to help prevent diabetes was to limit the rate of glucose absorption from the intestines into the bloodstream.

For the creation of therapeutic agents and to support the efficacy of *P. lentiscus’* chemical makeup as a nutraceutical preservative, research into the physicochemical qualities of potential compounds is essential. The pharmacokinetic and physicochemical characteristics of the discovered chemicals in the EO and aqueous extract (E_0_) of *P. lentiscus* leaves, as well as their similarity to medicines, were studied. For PASS and ADMET prediction analyses, the principal components of the EO and aqueous extract (E_0_) of *P. lentiscus* were chosen. According to the findings of the PASS and ADMET prediction studies, all compounds had significant values for a number of activities, including antioxidant, antifungal, antibacterial potential, coagulant effect, antithrombotic effect, hemostatic effect, inhibition of α-amylase, and inhibition of α-glucosidase.

According to our predictions, all significant compounds showed good antioxidant, hemostatic, and antifungal properties and an inhibitory effect on α-amylase. Additionally, these compounds also exhibited good antithrombotic, antibacterial properties, and an inhibitory effect on α-glucosidase, but had a low coagulant effect. These results are consistent with previous studies that have demonstrated the antioxidant, antimicrobial, anticoagulant, and anti-diabetic properties of compounds present in *P. lentiscus* [105,106,107]. These findings suggest that the identified compounds in the EO and aqueous extract (E_0_) of *P. lentiscus* could be used as therapeutic agents for various diseases.

Phenolic acids and flavonoids, such as gallic acid, quinic acid, myricetin derivatives, and their derivatives, are known to be effective and long-lasting natural antioxidants that can trap free radicals. These compounds have therapeutic properties that can help treat diseases such as infections, diabetes, cardiovascular diseases, and cancer. Clinical trials are underway to evaluate the antibacterial, antifungal, and anticoagulant effects of these compounds. Gallic acid is a phenolic acid that is easily absorbed by the body. It can be absorbed in the stomach, small intestine, or both and then is transformed into methylated and glucuronidated forms in the plasma and urine. Flavonoids are a group of polyphenols containing more than six thousand compounds, mainly bound to β-glycosides in plants. Flavonoids bound to glucose, arabinose, or xylose are hydrolyzed by human cytosolic β-glucosidase [108,109].

Gallic acid has shown increasing beneficial effects against various metabolic disorders in clinical trials. In addition to reducing hyperglycemia and/or excessive lipid storage, it could improve inflammation and oxidative stress at the cellular level by modulating the expression of cytokines and enzymes such as SOD, δ-aminolevulinic acid dehydratase, CAT, GST, GSH, GPx, and GRd. The available results are generating great interest from the research and clinical community regarding the therapeutic potential of gallic acid for metabolic diseases [110,111,112,113,114,115,116].

Pharmacological studies in vitro and in vivo have shown that quinic acid and its derivatives have various biological activities, such as antioxidant, antimicrobial, antidiabetic, anticancer, antiviral, anti-aging, protective, antinociceptive, and analgesic effects. Quinic acid derivatives have demonstrated promising effects in the treatment of glycemic control, obesity, and dyslipidemia, as well as insulin secretion, among others. The results of clinical trials (ClinicalTrials.gov; ID: NCT02621060) show that quinic acid derivatives have excellent potential for glucose control, insulin secretion, and insulin sensitivity. Therefore, compounds such as 3-Galloylquinic acid, 3-5-Di-*O*-galloylquinic acid, and 3,4,5-Tri-*O*-galloylquinic acid could be a potential source of antidiabetic drugs [117].

Clinical studies have shown that myricetin derivatives have promising potential for usage in the medical field. For instance, it has been noted that the myricetin-containing supplement EmulinTM can lower blood glucose rise in people with type 2 diabetes [118]. Blueberin, which includes 50 mg of myricetin, was shown to reduce fasting blood glucose levels and serum C-reactive protein levels in a four-week randomized, controlled clinical experiment [119].

Other investigations have also reported a correlation between myricetin consumption and a reduced risk of cancer in men [120,121]. Given the strong link between inflammation, diabetes, and cancer with nervous system disorders, these previous clinical studies provide indirect evidence in favor of the possibility of developing clinical trials for the use of myricetin in the treatment of neurological disorders.

Molecular docking studies are commonly used to predict the interaction between a ligand and a protein, providing insights into the biological activity of natural sources. They also offer additional information on the interaction and potential mechanisms of action at the binding site of different proteins [122]. To predict these potential mechanisms, the major compounds of the EO and aqueous extract (E_0_) of *P. lentiscus* were subjected to docking studies. The proteins NADPH oxidase, thrombin, α-amylase, and α-glucosidase were used as receptors. This study demonstrated that constituents of *P. lentiscus* can modulate the proteolytic activity of thrombin, as well as the inhibitory activity of α-amylase and α-glucosidase, which are crucial in managing hyperglycemia. Indeed, among the studied compounds, some exhibited promising binding affinity with NADPH oxidase, thrombin, α-amylase, and α-glucosidase. Compounds with lower molecular weight, reduced lipophilicity, and lower hydrogen bonding capacity may have better permeability [123], good absorption, and bioavailability [124,125]. According to this hypothesis, most of the major compounds of *P. lentiscus* studied adhere to Lipinski’s rules, making them a potential source of natural nutraceutical preservatives.

## 5. Conclusions

Overall, the results obtained revealed the strong in vitro and in silico antioxidant power of *P. lentiscus* leaf extracts originating from the Boulemane region, as well as their interesting natural nutraceutical preserving properties for the agri-food industry and even the pharmaceutical and cosmetic industries, thanks to its proven in vitro, in vivo, and in silico antimicrobial, anticoagulant, and antidiabetic effects. The bioactive molecules in *P*. *lentiscus* leaf extracts can neutralize free radicals by giving an electron or hydrogen atom to stabilize them, which helps prevent oxidative damage to cells and tissues. The identified substances can also chelate or bind to metal ions, reducing their ability to initiate oxidative reactions. According to the results of in vitro studies (DPPH, FRAP, and TAC) as well as in silico prediction, the identified molecules in *P. lentiscus* extracts such as Myricetin-3-*O*-xyloside, 3,4,5-Tri-*O*-galloylquinic acid, 3-5-Di-*O*-galloylquinic acid, 3-Galloylquinic acid, and Gallic acid 3-*O*-gallate have a strong ability to neutralize harmful free radicals, regulate the production of reactive oxygen species (ROS), and reduce oxidative damage. Our results highlight the importance of *P. lentiscus* leaves as a potential source of beneficial health substances that can be used as natural alternatives to synthetic preservatives currently used.

## Figures and Tables

**Figure 1 biomedicines-11-02372-f001:**
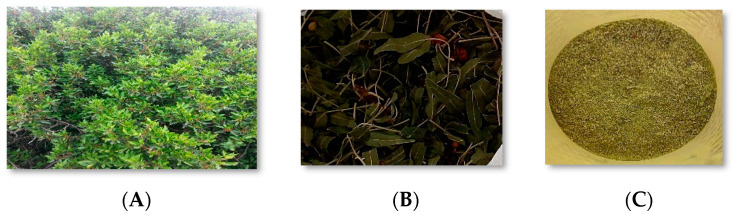
*Pistacia lentiscus* (L.); (**A**): Tree, (**B**): Dried leaves and fruits, (**C**): Powder of dried leaves. (Aziz Drioiche and Touriya Zair, 2021).

**Figure 2 biomedicines-11-02372-f002:**
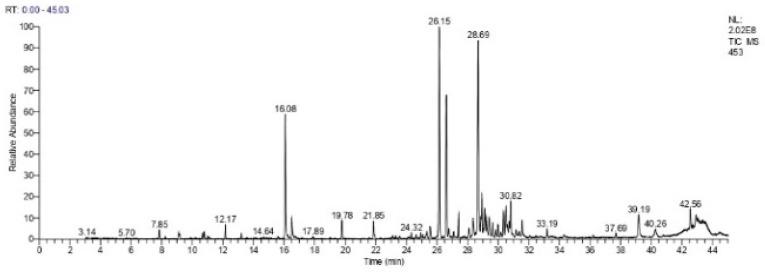
Chromatographic profile of the GC-MS analysis of the *P. lentiscus* EO studied.

**Figure 3 biomedicines-11-02372-f003:**
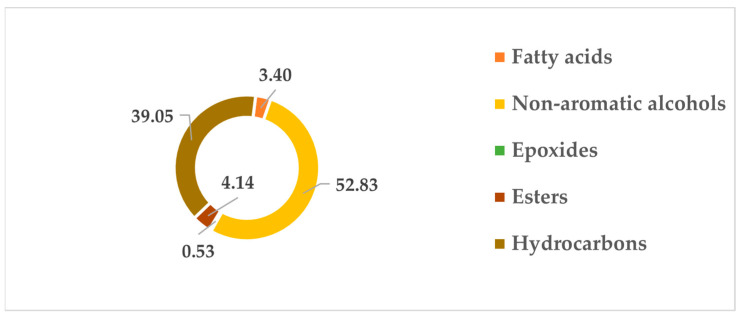
Distribution of chemical families identified in *P. lentiscus* EO.

**Figure 4 biomedicines-11-02372-f004:**
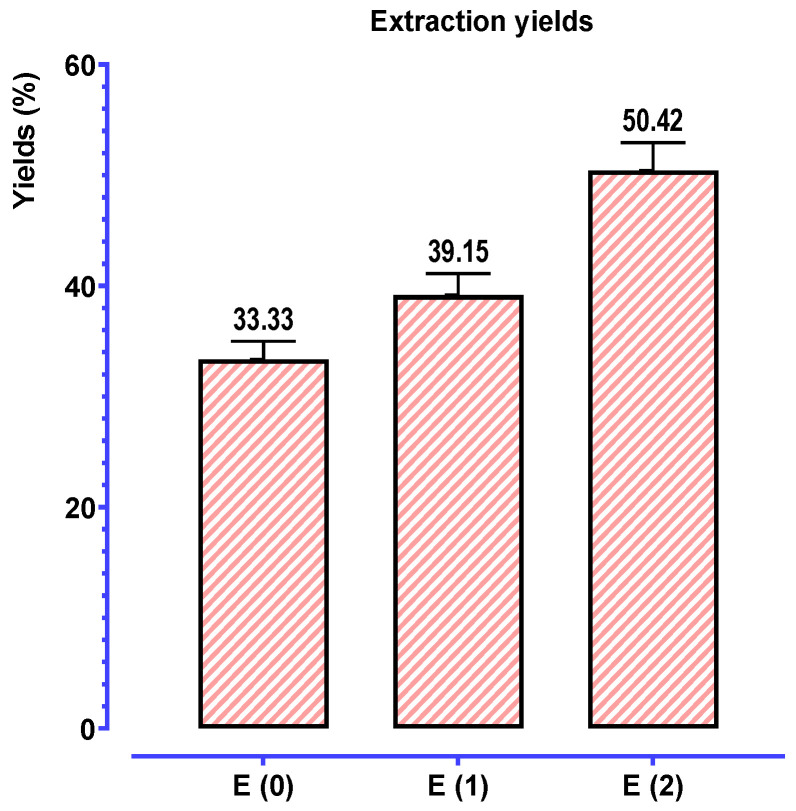
Extraction yields of phenolic compounds from *P. lentiscus*.

**Figure 5 biomedicines-11-02372-f005:**
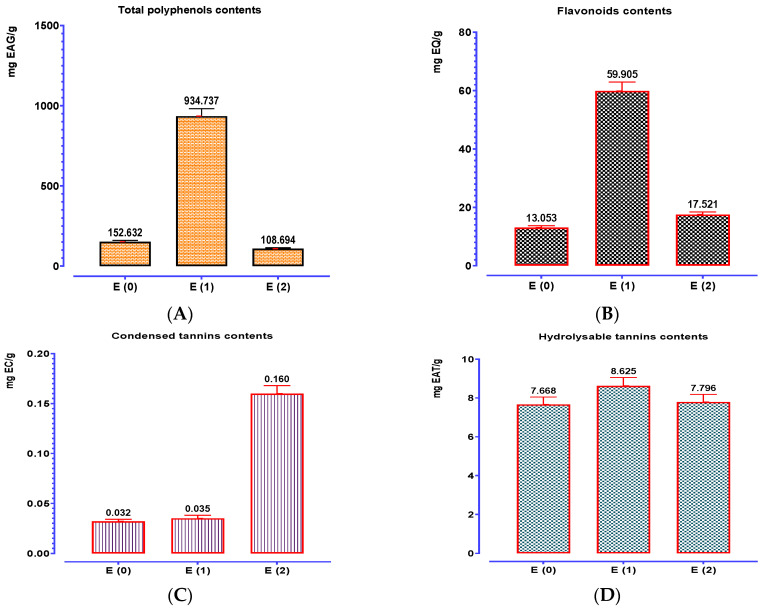
Contents of total polyphenols (**A**), flavonoids (**B**), condensed tannins (**C**), and hydrolyzable tannins (**D**); Mean values ± standard deviations of determinations performed in triplicate are reported; Means are significantly different (*p* < 0.001).

**Figure 6 biomedicines-11-02372-f006:**
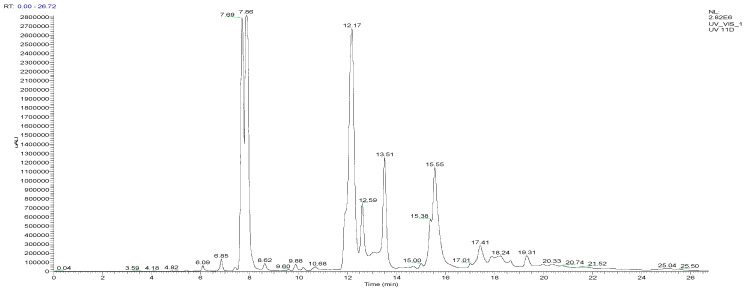
HPLC chromatograms of *P. lentiscus* compounds of aqueous extract (E_0_).

**Figure 7 biomedicines-11-02372-f007:**
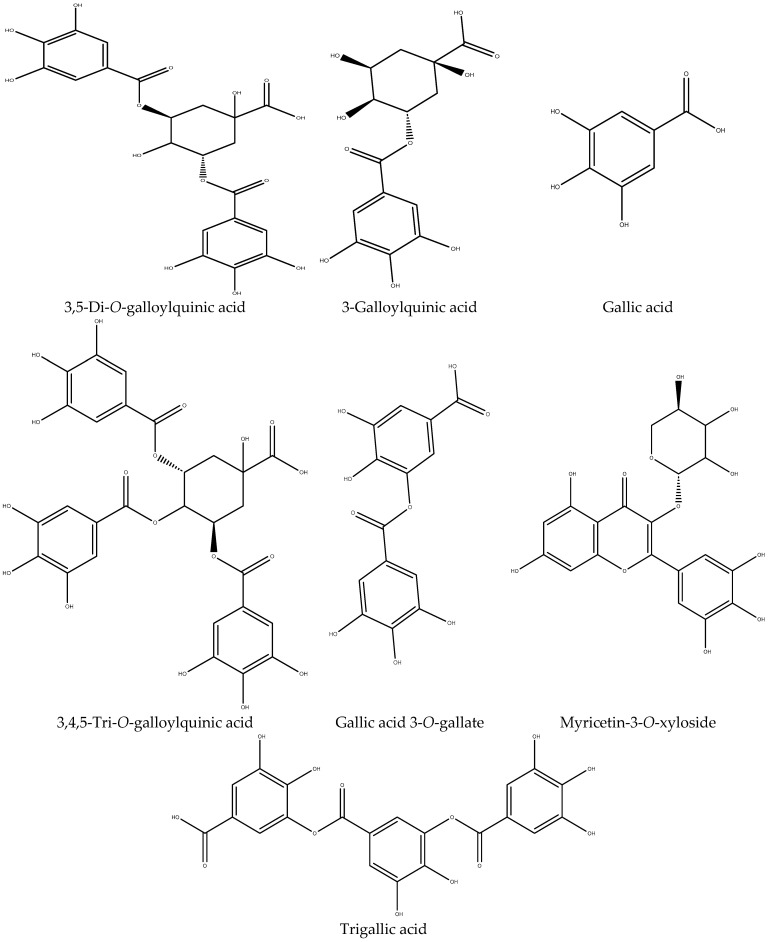
Structures of the majority of compounds identified in the aqueous extract E (0) of *P. lentiscus*.

**Figure 8 biomedicines-11-02372-f008:**
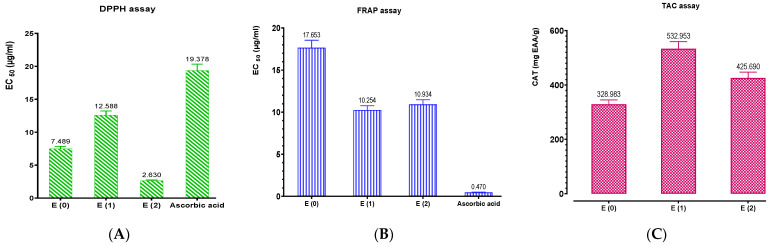
Antioxidant activity of ascorbic acid and extracts by DPPH (**A**), FRAP (**B**), and TAC (**C**) assays. Mean values ± standard deviations of determinations performed in triplicate are reported; Means are significantly different (*p* < 0.001).

**Figure 9 biomedicines-11-02372-f009:**
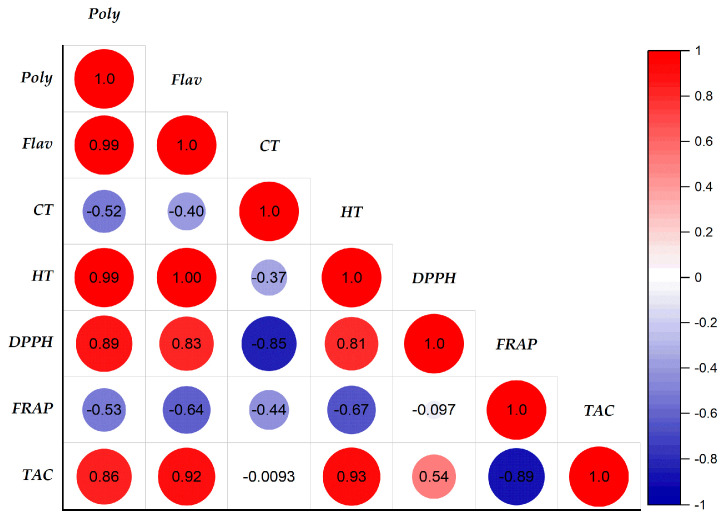
Correlation between antioxidant activity and phenolic compound content of *P. lentiscus* leaf extracts. (Poly: polyphenols; Flav: flavonoids; CT: condensed tannins; HT: hydrolyzable tannins).

**Figure 10 biomedicines-11-02372-f010:**
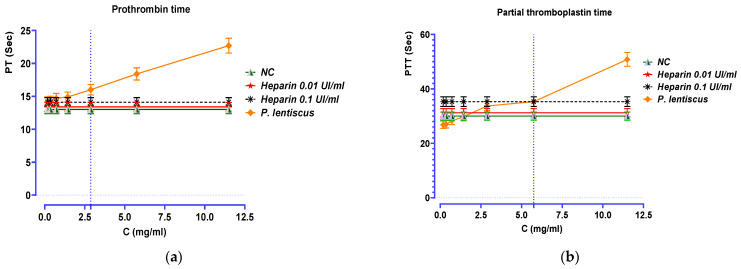
Effect of *P. lentiscus* decocted extract (E_0_), normal control (NC), and heparin on prothrombin time (**a**) and partial thromboplastin time (**b**).

**Figure 11 biomedicines-11-02372-f011:**
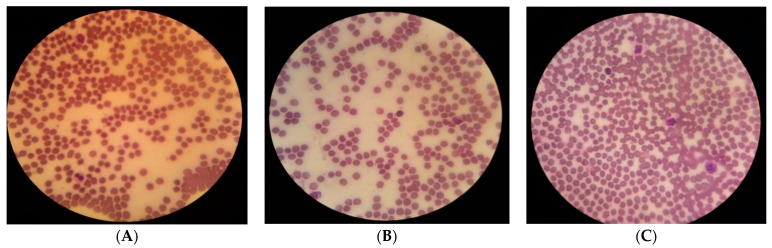
Blood smears of evaluated samples (100×), (**A**) Untreated blood sample, (**B**) Blood sample treated with aqueous extract (E_0_) at 11,500 mg/mL, (**C**) Blood sample treated with aqueous extract (E_0_) at 0.179 mg/mL.

**Figure 12 biomedicines-11-02372-f012:**
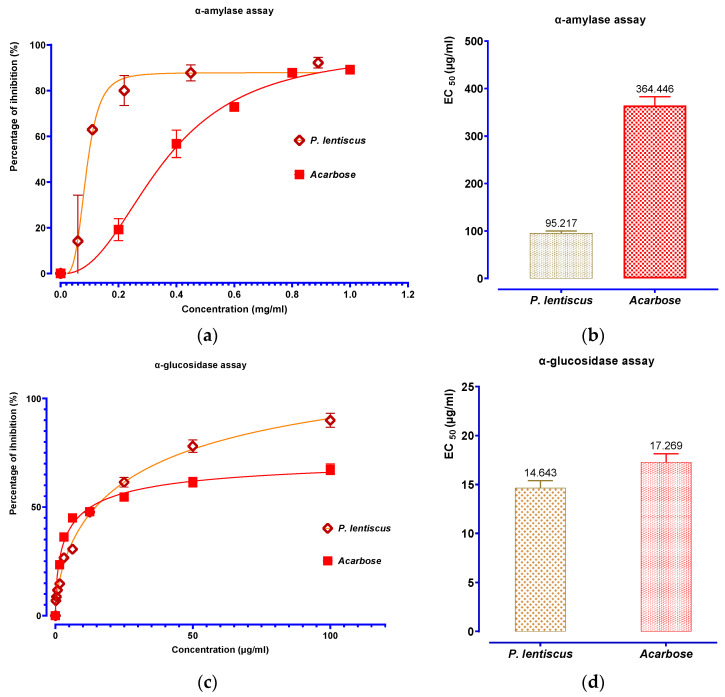
Percent inhibition and EC_50_ of inhibitory effects on α-amylase (**a**,**b**) and α-glucosidase (**c**,**d**) activities by *P. lentiscus* decocted extract and acarbose, in vitro. Values are means ± SEM (n = 3).

**Figure 13 biomedicines-11-02372-f013:**
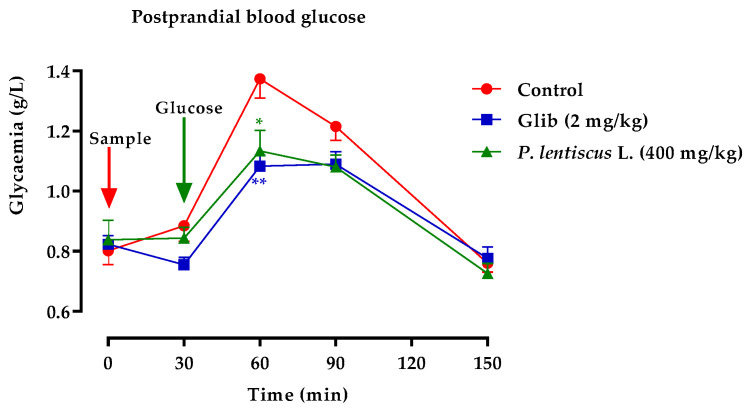
Variation in postprandial blood glucose in normal rats after administration of test products (decocted extract and glibenclamide). Values are means ± SEM. (n = 6). * *p* < 0.05, ** *p* < 0.01 in comparison with the control.

**Figure 14 biomedicines-11-02372-f014:**
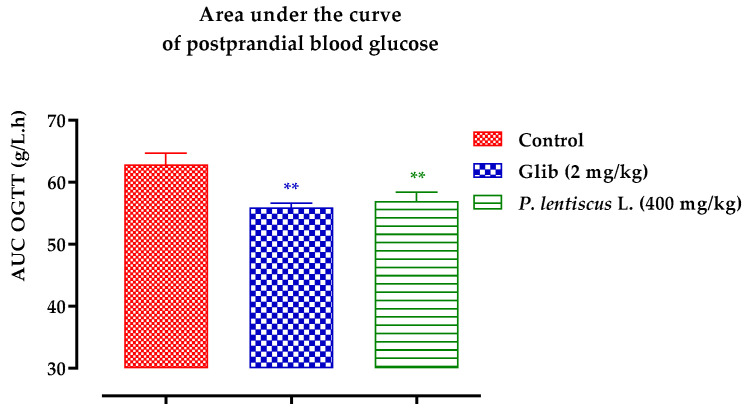
Variation in the area under the curve of postprandial blood glucose in normal rats after administration of tested products (decocted extract and glibenclamide). Values are means ± SEM. (n = 6). ** *p* < 0.01 when compared with control.

**Figure 15 biomedicines-11-02372-f015:**
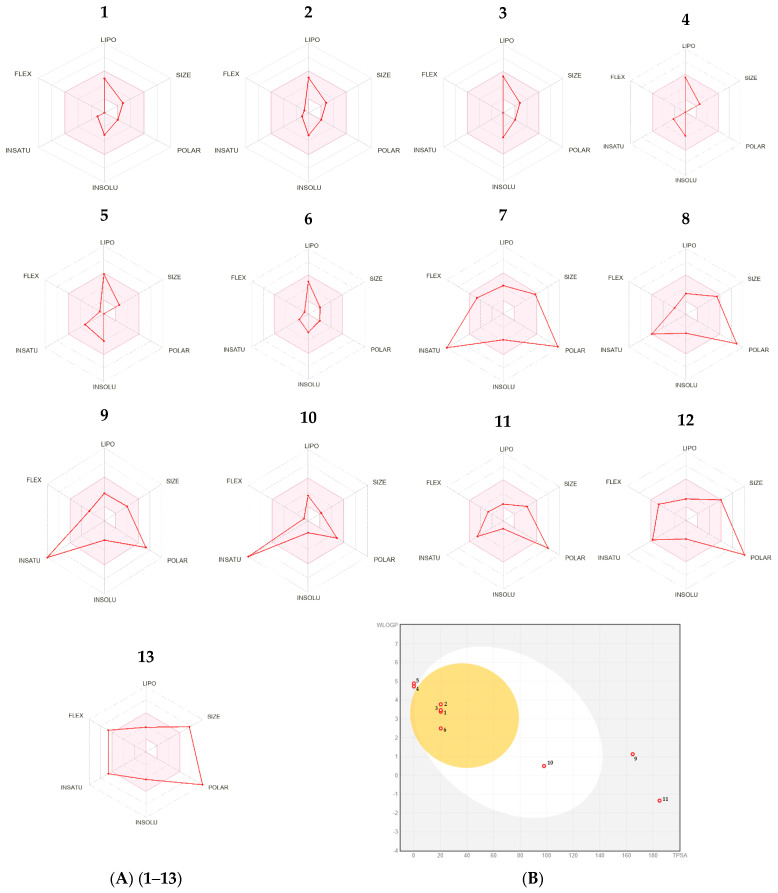
(**A**): bioavailability radar, (**B**): Predicted BOILED-Egg diagram of Spathulenol (**1**), α-Cadinol (**2**), Globulol (**3**), Bicyclogermacrene (**4**), Germacrene D (**5**), Terpinen-4-ol (**6**), Trigallic acid (**7**), Myricetin-3-*O*-xyloside (**8**), Gallic acid 3-*O*-gallate (**9**), Gallic acid (**10**), 3-Galloylquinic acid (**11**), 3-5-Di-*O*-galloylquinic acid (**12**); 3,4,5-Tri-*O*-galloylquinic acid (**13**).

**Figure 16 biomedicines-11-02372-f016:**
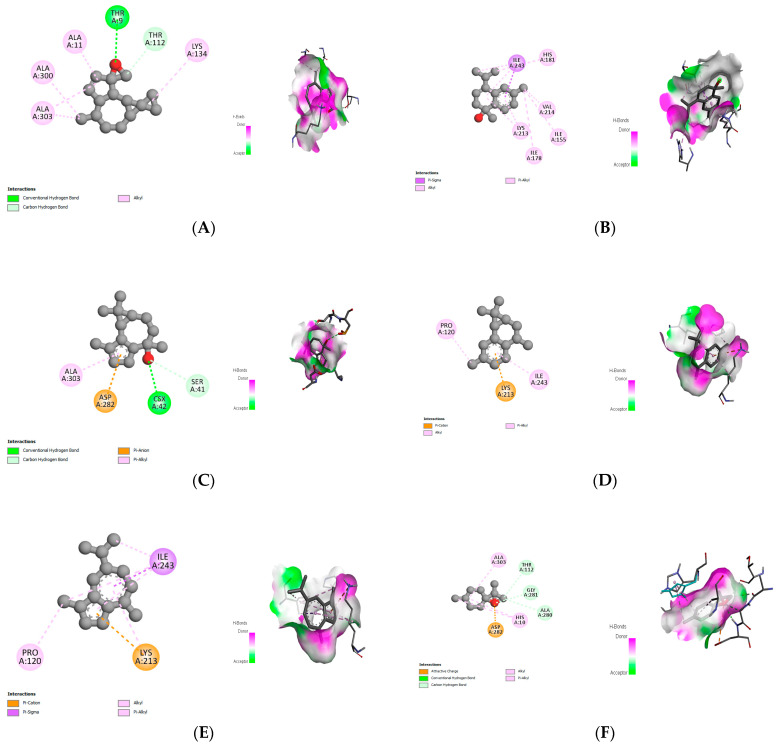
2D and 3D interaction of NADPH with ligands. (**A**) Spathulenol, (**B**) α-Cadinol, (**C**) Globulol, (**D**) Bicyclogermacrene, (**E**) Germacrene D and (**F**) Terpinen-4-ol.

**Table 1 biomedicines-11-02372-t001:** *P. lentiscus’s* origin, parts used, habitat, and season of harvest.

Scientific Name	Part Collected	Type of Extract Used	Harvesting Area						
			Region	Province	Municipality	Latitude (x)	Longitude (y)	Altitude (m)	Collection Period
*P. lentiscus* L.	Leaves	Essential oils and Extracts	Fez-Meknes	Boulemane	Oulad Ali Youssef	33°28′41″ N	3°59′35″ W	1885.5 m	21 December

**Table 2 biomedicines-11-02372-t002:** List of tested bacterial and fungal strains, together with references.

Strains		Abbreviations	References
Gram-positive cocci	*Staphyloccocus epidermidis*	*S. epidermidis*	5994
	*Staphyloccocus aureus BLACT*	*S. aureus BLACT*	4IH2510
	*Staphyloccocus aureus* STAIML/MRS/mecA/HLMUP/BLACT	*S*. *aureus* STAIML/MRS/mecA/HLMUP/ BLACT	2DT2220
	*Streptococcus acidominimus*	*S. acidominimus*	7DT2108
	*Streptococcus group D*	*S. group D*	3EU9286
	*Streptococcus agalactiae*	*S. agalactiae*	7DT1887
	*Streptococcus porcinus*	*S. porcinus*	2EU9285
	*Enterococcus faecalis*	*E. faecalis*	2CQ9355
	*Enterococcuss faecium*	*E. faecium*	13EU7181
Gram-negative bacilli	*Acinetobacter baumannii*	*A. baumannii*	7DT2404
	*Escherichia coli*	*E. coli*	3DT1938
	*Escherichia coli ESBL*	*E. coli ESBL*	2DT2057
	*Enterobacter aerogenes*	*E. aerogenes*	07CQ164
	*Enterobacter cloacae*	*E. cloacae*	02EV317
	*Citrobacter koseri*	*C. koseri*	3DT2151
	*Klebsiella pneumonie* ssp. *pneumonie*	*K. pneumonie*	3DT1823
	*Proteus mirabilis*	*P. mirabilis*	2DS5461
	*Pseudomonas aerogenosa*	*P. aerogenosa*	2DT2138
	*Pseudomonas fluorescence*	*P. fluorescence*	5442
	*Pseudomonas putida*	*P. putida*	2DT2140
	*Serratia marcescens*	*S. marcescens*	375BR6
	*Salmonella* sp.	*Salmonella* sp.	2CG5132
	*Shigella* sp.	*Shigella* sp.	7DS1513
	*Yersinia enterocolitica*	*Y. enterocolitica*	ATCC27729
Yeasts	*Candida albicans*	*C. albicans*	Ca
	*Candida kefyr*	*C. kefyr*	Cky
	*Candida krusei*	*C. krusei*	Ckr
	*Candida parapsilosis*	*C. parapsilosis*	Cpa
	*Candida tropicalis*	*C. tropicalis*	Ct
	*Candida dubliniensis*	*C. dubliniensis*	Cd
	*Saccharomyces cerevisiae*	*S. cerevisiae*	Sacc
Fungi	*Aspergillus niger*	*A. niger*	AspN

**Table 3 biomedicines-11-02372-t003:** Extraction coding.

Extraction Methods	Solvents	Codification
**Soxhlet**	Ethanol/Water (70/30; *v*/*v*)	E (2)
	Water	E (1)
**Decoction**	Water	E (0)

**Table 4 biomedicines-11-02372-t004:** Moisture content, pH, ash, and heavy metals in *P. lentiscus* leaves.

MC (%)	pH	Ash (%)	Heavy Metal Concentration (mg/L)						
			Arsenic (As)	Cadmium (Cd)	Chromium (Cr)	Iron (Fe)	Lead (Pb)	Antimony (Sb)	Titanium (Ti)
10.12 ± 0.51	3.52 ± 0.18	5.11 ± 0.26	0.1131	0.0435	0.0639	0.3479	0.0976	0.1175	0.0692

**Table 5 biomedicines-11-02372-t005:** The yield of the essential oil of *P. lentiscus*.

Properties	Yield (%)	Density	Color	Odor
*P. lentiscus*	0.22 ± 0.01	0.933	Transparent	Aromatic

**Table 6 biomedicines-11-02372-t006:** Chemical composition of *P. lentiscus* EO studied.

IK	Composés	Aire (%)
**939**	**Pinene <α->**	0.52
975	Sabinene	0.40
1026	Cymene <ο->	0.46
1059	Terpinene <γ->	0.93
**1177**	**Terpinen-4-ol**	**9.95**
1188	Terpineol <α->	1.86
1285	Bornyl acetate	1.33
1338	Elemene <δ->	1.46
1441	Aromadendrene	1.36
**1481**	**Germacrene D**	**17.54**
**1500**	**Bicyclogermacrene**	**12.52**
1500	Muurolene <α->	0.61
1513	Cadinene <γ->	0.46
1523	Cadinene <δ->	1.92
1550	Agarofuran <α->	0.87
1566	Hexenyl benzoate <(3Z)->	2.03
**1578**	**Spathulenol**	**18.57**
1590	Globulol	3.94
1602	Ledol	2.20
1595	Cubeban-11-ol	1.64
1600	Rosifoliol	1.55
1619	Junenol	0.67
1650	Eudesmol <β->	1.00
1652	Cedr-8(15)-en-10-ol	2.00
1654	Cadinol <α->	5.09
1663	Eudesmol <7-epi-α->	0.64
1676	Ledenoxide	0.53
1686	Germacra-4(15).5.10(14)-trien-1-α-ol	0.81
1760	Benzyl benzoate	0.78
1780	Muurolene <14-hydroxy-α->	1.38
1943	Phytol	1.53
1960	Hexadecanoic acid	2.22
2130	Linolenic acid	1.18
	**Hydrocarbon monoterpenes**	**2.31**
	**Oxygenated monoterpenes**	**15.17**
	**Hydrocarbon sesquiterpenes**	**35.87**
	**Oxygenated sesquiterpenes**	**45.07**
	**Oxygenated diterpenes**	**1.53**
	**Total**	**99.95**

**Table 7 biomedicines-11-02372-t007:** Results of phytochemical tests.

Compounds/Species		*P. lentiscus*
Part Used		Leaves
Sterols and triterpenes		++
Flavonoids		++
Anthocyanins		++
Leucoanthocyanins		+++
Tannins	Catechic tannins	+
	Gallic tannins	+++
Anthracene derivatives	Quinones	−
	O-Heterosides	+
	C-Heterosides	++
Saponosides		++
Monosaccharides and holosides		+++
Alkaloids	Dragendorff	−
	Mayer	−
Reducing Compounds		−
Mucilages		+

Category: Strong presence: +++; average presence: ++; low presence: + and absent: −.

**Table 8 biomedicines-11-02372-t008:** List of compounds identified by mass spectrometry in *P. lentiscus* leaf aqueous extract E (0).

N°	RT	Molecules	Structure	Classes	Exact Masses	[M-H]^−^ (*m*/*z*)	Fragment Ions (*m*/*z*)	Area (%)
1	3.59	Quinic acid	C_7_H_12_O_6_	Phenolic acid	192	191	191, 127	0.01
2	4.18	Gallocatechin	C_15_H_14_O_7_	Flavonoid	306	305	305, 179, 261, 125	0.01
3	4.82	Shikimic acid	C_7_H_10_O_5_	Other	174	173	173, 137, 111	0.02
4	5.42	Kaempferol pentoside	C_20_H_18_O_10_	Flavonoid	418	417	417, 284, 255	0.03
5	6.09	Myricetin	C_15_H_10_O_8_	Flavonoid	318	317	317, 137	0.26
6	6.85	Β-Glucogallin	C_13_H_16_O_10_	Polyphenol	332	331	331, 169, 151	0.56
7	7.69	Gallic acid	C_7_H_6_O_5_	Phenolic acid	170	169	169, 125	13.61
8	7.87	3-Galloyl quinic acid	C_14_H_16_O_10_	Phenolic acid	344	343	343, 191	18.01
9	8.62	Epigallocatechin gallate	C_22_H_18_O_11_	Flavonoid	458	457	457, 125	0.45
10	9.60	Apigenin galloyl glucoside	C_29_H_44_O_12_	Flavonoid	584	583	583, 431, 269	0.6
11	9.88	Galloyl shikimic acid	C_14_H_14_O_9_	Polyphenol	326	325	325, 169, 151, 125	0.31
12	10.68	1,5-Di-*O*-galloyl quinic acid	C_21_H_20_O_14_	Phenolic acid	496	495	495, 343, 191	0.38
13	10.89	Gentisic acid	C_7_H_6_O_4_	Phenolic acid	154	153	153, 109	0.04
14	11.37	Luteolin	C_15_H_10_O_6_	Flavonoid	286	285	285, 217, 175, 151	0.06
15	12.17	3,5-Di-*O*-galloyl quinic acid	C_21_H_20_O_14_	Phenolic acid	496	495	495, 343, 247, 191	26.41
16	12.59	Myricetin-3-*O*-xyloside	C_20_H_18_O_12_	Flavonoid	450	449	449, 316	4.91
17	13.06	Phloretin-2-*O*-glucoside	C_21_H_24_O_10_	Flavonoid	436	435	435, 273, 167	2.03
18	13.51	Gallic acid 3-*O*-gallate	C_14_H_10_O_9_	Phenolic acid	322	321	321, 169, 125	8.23
19	15.00	Quercetin pentoside	C_20_H_18_O_11_	Flavonoid	434	433	433, 301, 255, 242, 193	0.87
20	15.38	Caffeic acid 3-glucoside	C_15_H_18_O_9_	Phenolic acid	342	341	341, 179, 161, 135	2.76
21	15.55	3,4,5-Tri-*O*-galloyl quinic acid	C_28_H_24_O_18_	Phenolic acid	648	647	647, 495, 343, 169	11.13
22	17.01	Luteolin-7-*O*-glucuronide	C_21_H_18_O_12_	Flavonoid	462	461	461, 285	0.29
23	17.41	Trigallic acid	C_21_H_14_O_13_	Phenolic acid	474	473	473, 321, 169	3.84
24	18.24	Myricetin-3-*O*-glucoside	C_21_H_20_O_13_	Flavonoid	480	479	479, 317, 271, 179	2.53
25	19.31	Rutin	C_27_H_30_O_16_	Flavonoid	610	609	609, 301	1.14
26	20.33	1,3,4,5-tetra-*O*-galloylquinic acid	C_35_H_28_O_22_	Phenolic acid	800	799	647, 495, 343, 191	0.95
27	20.74	Myricetin-3-*O*-rhamnoside	C_21_H_20_O_12_	Flavonoid	464	463	463, 317	0.2
28	21.52	Myricetin galloyl rhamnopyranoside	C_28_H_24_O_16_	Flavonoid	616	615	615, 463, 301	0.02
29	22.24	Tetragalloyl hexose	C_34_H_28_O_22_	Polyphenol	788	787	787, 617, 465	0.01
30	25.04	Quercetin	C_15_H_10_O_7_	Flavonoid	302	301	301, 151	0.22
31	25.83	Myricetin galloyl hexoside	C_28_H_24_O_17_	Flavonoid	632	631	631, 479, 317	0.02
Total								99.91

**Table 9 biomedicines-11-02372-t009:** The MIC, MBC, and MFC (mg/mL) of *P. lentiscus* decocted extract and EO, and the MIC (μg/mL) of antibiotics and antifungal agent.

Microorganism		EO		Extract		Antibiotics *					Antifungals ^#^
		MIC	MBC or MFC	MIC	MBC or MFC	Gentamicin	Amoxicillin–Clavulanate	Vancomycin	Trimethoprim-Sulfamethoxazole	Penicillin G	Terbinafine
** *GPC* **	** *S.* *epidermidis* **	5	5	2.5	2.5	2		>8	>4/76		
	** *S.* *aureus BLACT* **	5	5	1.2	1.2	<0.5		2	<10		
	** *S.* *aureus STAIML/MRS/mecA/HLMUP/BLACT* **	>5	>5	2.5	2.5	2		>8	>4/76		
	** *S.* *acidominimus* **	>5	>5	2.5	5	≤250		<0.5		0.03	
	** *S.* *group D* **	5	5	1.2	1.2	>1000		<0.5		0.13	
	** *S.* *agalactiae (B)* **	>5	>5	1.2	1.2	≤250		>4		0.06	
	** *S.* *porcinus* **	>5	>5	2.5	2.5	≤250		<0.5		0.06	
	** *E.* *faecalis* **	>5	>5	2.5	2.5	≤500		1	≤0.5/9.5		
	** *E.* *faecium* **	>5	>5	1.2	1.2	≤500		>4	>4/76		
** *GNB* **	** *A.* *baumannii* **	>5	>5	1.2	2.5	≤1	≤2/2		≤1/19		
	** *E.* *coli* **	5	5	2.5	2.5	2	8/2		≤1/19		
	** *E.* *coli* ** ** *ESBL* **	5	5	2.5	5	2	>8/2		>4/76		
	** *E.* *aerogenes* **	>5	>5	1.2	1.2	≤1	8/2		≤1/19		
	** *E.* *cloacae* **	>5	>5	1.2	2.5	>4	>8/2		>4/76		
	** *C.* *koseri* **	2.5	5	2.5	2.5	<1	>8/2		<20		
	** *K.* *pneumoniae* **	>5	>5	0.6	1.2	≤1	≤2/2		≤1/19		
	** *P.* *mirabilis* **	>5	>5	2.5	2.5	2	≤2/2		>1/19		
	** *P.* *aeruginosa* **	5	5	2.5	2.5	2	>8/2		4/76		
	** *P.* *fluorescence* **	>5	>5	1.2	1.2	4	>8/2		4/76		
	** *P.* *putida* **	5	5	2.5	5	>4	>8/2		>4/76		
	** *S.* *marcescences* **	>5	>5	2.5	2.5	4	>8/2		>4/76		
	***Salmonella* sp.**	5	5	1.2	2.5	>4	8/2		>4/76		
	***Shigella* sp.**	2.5	2.5	1.2	1.2	>4	8/2		>4/76		
	** *Y.* *enterolitica* **	>5	>5	1.2	1.2	≤1	8/2		2/38		
** *Yeasts* **	** *C.* *albicans* **	5	5	1.2	1.2						12,500
	** *C.* *kefyr* **	>5	>5	2.5	2.5						25,000
	** *C.* *krusei* **	>5	>5	0.3	0.6						50,000
	** *C.* *parapsilosis* **	5	5	2.5	2.5						6250
	** *C.* *tropicalis* **	>5	>5	5	5						12,500
	** *C.* *dubliniensis* **	5	5	5	5						3125
	** *S.* *cerevisiae* **	>5	>5	5	5						3125
** *Molds* **	** *A. niger* **	>5	>5	2.5	5						3125

*: the MIC (μg/mL) of the antibiotics was determined by the BD Phoenix™ identification and antibiogram instrument; #: the MIC (μg/mL) of terbinafine was determined on a microplate.

**Table 10 biomedicines-11-02372-t010:** Results of blood counts for samples studied.

Blood Count				Decocted Concentration (mg/mL)				
Parameters	Control Sample	0.179	0.359	0.719	1.438	2.875	5.75	11.5
**Leukocytes**	6.5 ± 0.3	6.2 ± 0.3	6.2 ± 0.3	6.4 ± 0.3	6.4 ± 0.3	6.3 ± 0.3	6.2 ± 0.3	5.7 ± 0.3
**Red blood cells**	4.37 ± 0.2	4.15 ± 0.2	4.22 ± 0.2	4.19 ± 0.2	4.22 ± 0.2	4.22 ± 0.2	4.22 ± 0.2	4.01 ± 0.2
**Hemoglobin**	12.9 ± 0.6	12.1 ± 0.6	12.3 ± 0.6	12.3 ± 0.6	12.3 ± 0.6	12.4 ± 0.6	12.4 ± 0.6	11.7 ± 0.6
**Hematocrit**	38.4 ± 1.9	36.3 ± 1.8	37.3 ± 1.9	37.0 ± 1.9	37.3 ± 1.9	37.4 ± 1.9	37.3 ± 1.9	35.6 ± 1.8
**Platelets**	239 ± 12.0	265 ± 13.3	260 ± 13.0	263 ± 13.2	252 ± 12.6	275 ± 13.8	278 ± 13.9	252 ± 12.6
**Mean platelet volume**	9.4 ± 0.5	9.2 ± 0.5	9.1 ± 0.5	9.3 ± 0.5	9.2 ± 0.5	9.2 ± 0.5	9.1 ± 0.5	9.4 ± 0.5
**Platelet distribution index**	14.7 ± 0.7	14.6 ± 0.7	14.6 ± 0.7	14.8 ± 0.7	14.6 ± 0.7	14.8 ± 0.7	14.8 ± 0.7	14.8 ± 0.7

**Table 11 biomedicines-11-02372-t011:** In silico PASS, ADMET, and predictive toxicity analysis (Pro-Tox II) of EO and decocted major metabolites of *P. lentiscus*.

Prediction	Parameters	EO 1	EO 2	EO 3	EO 4	EO 5	EO 6	D 1	D 2	D 3	D 4	D 5	D 6	D 7
PASS Prediction (Pa/Pi)														
Antioxidant	Antioxidant						0.151/0.102	0.516/0.006	0.944/0.002	0.516/0.006	0.520/0.006	0.791/0.003	0.784/0.004	0.788/0.003
Antimicrobial	Antifungal	0.519/0.027	0.454/0.039	0.484/0.033	0.439/0.042	0.570/0.022	0.466/0.036	0.421/0.045	0.711/0.009	0.421/0.045	0.398/0.050	0.564/0.022	0.528/0.026	0.513/0.028
	Antibacterial	0.408/0.028	0.444/0.023	0.381/0.035	0.332/0.049	0.427/0.025	0.328/0.050	0.199/0.015	0.585/0.010	0.437/0.023	0.418/0.026	0.516/0.015	0.490/0.017	0.488/0.018
Anticoagulant	Coagulant	0.135/0.049	0.157/0.035	0.145/0.042	0.143/0.043		0.208/0.015	0.192/0.020		0.192/0.020	0.253/0.008	0.105/0.079	0.116/0.066	0.128/0.055
	Antithrombotic	0.224/0.174	0.363/0.069	0.230/0.166	0.380/0.061	0.339/0.080	0.240/0.155	0.559/0.020	0.615/0.014	0.559/0.020	0.504/0.029	0.308/0.095	0.360/0.070	0.262/0.132
	Hemostatic	0.169/0.150	0.174/0.141	0.191/0.113		0.173/0.143	0.213/0.085	0.582/0.004	0.939/0.002	0.582/0.004	0.467/0.007	0.310/0.024	0.339/0.0180	0.282/0.034
Anti-diabetic	α-amylase inhibitor						0.177/0.069	0.475/0.008	0.148/0.091	0.475/0.008	0.522/0.005	0.368/0.020	0.401/0.016	0.302/0.030
	α-glucosidase inhibitor	0.086/0.042	0.076/0.054	0.087/0.041	0.090/0.038	0.079/0.050	0.096/0.033	0.136/0.013	0.776/0.001	0.136/0.013	0.153/0.010	0.203/0.005	0.188/0.006	0.173/0.007
ADME Prediction														
Physiochemical Properties	TPSA (Å^2^)	20.23	20.23	20.23	0.00	0.00	20.23	231.51	210.51	164.75	97.99	184.98	251.74	318.50
	Molar Refractivity	68.34	70.72	68.82	68.78	70.68	48.80	110.36	106.21	74.92	39.47	75.81	111.52	147.23
Drug Likeness Prediction	Bioavailability Score	0.55	0.55	0.55	0.55	0.55	0.55	0.11	0.17	0.11	0.56	0.11	0.11	0.11
	Synthetic accessibility	3.78	4.29	3.58	4.34	4.55	3.28	2.99	5.09	2.45	1.22	3.84	4.37	4.99
Absorption Parameters Prediction	Water solubility	−3.866	−4.073	−4.313	−5.364	−5.682	−2.296	−2.892	−2.892	−2.875	−2.56	−2.589	−2.895	−2.892
	Caco2 permeability	1.388	1.479	1.483	1.415	1.436	1.502	−1.699	−0.973	−1.279	−0.081	−0.997	−1.465	−1.909
	Intestinal absorption (human)	93.235	94.296	92.814	95.014	95.59	94.014	46.046	42.509	45.971	43.374	22.529	23.584	0
	Skin Permeability	−2.141	−1.923	−2.178	−1.658	−1.429	−2.182	−2.735	−2.735	−2.735	−2.735	−2.735	−2.735	−2.735
	P-glycoprotein substrate	No						Yes	Yes	Yes	No	Yes	Yes	Yes
	P-glycoprotein I inhibitor	No												
	P-glycoprotein II inhibitor	No												
Distribution Parameters Prediction	VDss (human)	0.522	0.42	0.556	0.648	0.544	0.21	0.348	1.549	-0.213	−1.855	1.039	1.557	1.014
	Fraction unbound (human)	0.326	0.28	0.259	0.298	0.261	0.514	0.198	0.184	0.3	0.617	0.64	0.289	0.287
	BBB permeability	0.6	0.596	0.632	0.716	0.723	0.563	−2.466	−1.789	−1.761	−1.102	−1.679	−2.63	−3.487
	CNS permeability	−2.447	−2.151	−2.176	−2.332	−2.138	−2.473	−4.424	−4.435	−3.855	−-3.74	−4.166	−4.848	−5.279
Metabolism Parameters Prediction	CYP2D6 substrate	No				No								
	CYP3A4 substrate	Yes	No	Yes	No									
	CYP1A2 inhibitor	No	No	Yes	Yes									
	CYP2C19 inhibitor	Yes	No	No	No									
	CYP2C9 inhibitor	No												
	CYP2D6 inhibitor													
	CYP3A4 inhibitor													
Excretion	Total Clearance	0.895	1.085	0.817	1.09	1.42	1.269	0.432	0.303	0.461	0.518	0.594	0.552	0.529
	Renal OCT2 substrate	No												
Toxicity	AMES toxicity	No												
	hERG I inhibitors													
	Skin Sensitization	Yes						No						
	LD_50_ (mg/kg)	3900	2830	2000	5300	5300	1016	2260	NA	2260	2000	3700	3700	3700
	Predicted Toxicity Class	5	5	4	5	5	4	5	NA	5	4	5	5	5
	Hepatotoxicity	Inactive												
	Carcinogenicity	Inactive	Inactive	Inactive	Inactive	Inactive	Inactive	Inactive	Inactive	Inactive	Active (0.56%)	Inactive	Inactive	Inactive
	Immunotoxicity	Inactive	Active (0.69%)	Inactive	Inactive	Active (0.80%)	Inactive	Inactive	Active (0.96%)	Inactive	Inactive	Active (0.63%)	Inactive	Inactive
	Mutagenicity	Inactive												
	Cytotoxicity													

Where “Pa” is probable activity, “Pi” is probable inactivity, “Å^2^” polar surface area, and Spathulenol (EO1), α-Cadinol (EO2), Globulol (EO3), Bicyclogermacrene (EO4), Germacrene D (EO5), Terpinen-4-ol (EO6), Trigallic acid (D1), Myricetin-3-*O*-xyloside (D2), Gallic acid 3-*O*-gallate (D3), Gallic acid (D4), 3-Galloylquinic acid (D5), 3-5-Di-*O*-galloylquinic acid (D6); 3,4,5-Tri-*O*-galloylquinic acid (D7).

**Table 12 biomedicines-11-02372-t012:** Details of interacting residues and binding affinities.

	Macromolecules	NADPH		Thrombin		α-Amylase		α-Glucosidase	
N°	Ligands	Binding Affinity	Residues H-Bonding	Binding Affinity	Residues H-Bonding	Binding Affinity	Residues H-Bonding	Binding Affinity	Residues H-Bonding
1	Spathulenol	−8.2	THR9						
2	α-Cadinol	−7.5							
3	Globulol	−7.9	CSX42						
4	Bicyclogermacrene	−7.7							
5	Germacrene D	−7.3							
6	Terpinen-4-ol	−6.4							
7	Trigallic acid	−11.2	HIS10, ASP282, THR112, LYS134, GLU32; HIS79	−7.8	LEU40; LEU41; GLU192; TRP141	−8.4	ASP356; GLY304; ARG303; ASP353; HIS305; 3L9503	−7.7	PHE680; THR790; GLU301; ARG699
8	Myricetin-3-*O*-xyloside	−8.6	CSX42; ASN36; ASP282; LYS134; THR112	−7.5	SER195; LEU41; GLY193; ARG35; TRP60D	−8.1	GLY9; ARG10; SER289; GLY334; ARG252; PRO332; ASP290; ARG291	−7.2	ARG699; THR681; GLU301
9	Gallic acid 3-*O*-gallate	−9.7	GLY12; HIS10; LYS134; SER41; 1SP282	−6.5	TRP60D; GLY193; SER195; LEU40	−7.5	ARG10; THR6; ARG252; ARG421; GLY334	−7.2	PHE680; GLU301; ASP305; ARG670
10	Gallic acid	−7.2	CSX42; SER41; HIS10; ALA11; ASP282	−4.5	S491251	−5.5	ARG421; GLY403; ARG398; ARG252; SER289	−5.0	ASN668; ILE803; ASP802; MET801; ARG536
11	3-Galloylquinic acid	−9.8	THR9; THR112; GLY12; SER41; LYS134; ASP282; HIS10	−6.6	CYS42; S491251; ASN143; GLN151	−7.5	SER3; ARG252; SER289; ARG421; GLY403; ASP402; GLY334; ARG10	−7.3	PHE680; GLU301; ASP305; ARG670
12	3-5-Di-*O*-galloylquinic acid	−10.0	SER41; LYS134; SER115; ASN36; ASN34; ASP282	−8.1	SER195; LEU41; TRP141; ASN143; ASP60E	−7.6	PRO332; ARG252; GLN7	−8.1	ARG699; PHE680; ASP305; ARG670; SO41010
13	3,4,5-Tri-*O*-galloylquinic acid	−10.4	GLU32; CSX42; SER41; LYS134; HIS10; ASP282; ASN248	−7.4	SER195; TRP60D; ASP60E; ASN143; GLU192	−8.4	GLN7; ARG10; GLY9; ASP402; THR11; ARG398; ARG252; THR6	−8.1	THR681; ARG814; ASP305

**Table 13 biomedicines-11-02372-t013:** 2D and 3D interaction of NADPH, thrombin, α-amylase, and α-glucosidase with ligands. (**A**) Trigallic acid, (**B**) Myricetin-3-*O*-xyloside, (**C**) Gallic acid 3-*O*-gallate, (**D**) Gallic acid, (**E**) 3-Galloylquinic acid, (**F**) 3-5-Di-*O*-galloylquinic acid, and (**G**) 3,4,5-Tri-*O*-galloylquinic acid.

	NADPH	Thrombin	α-Amylase	α-Glucosidase
(**A**)	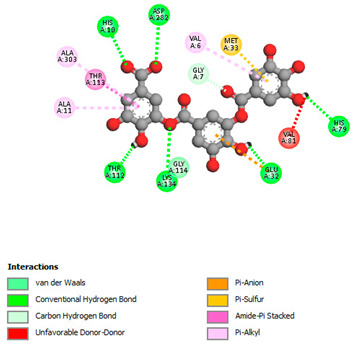	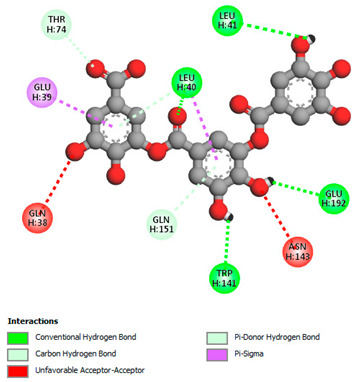	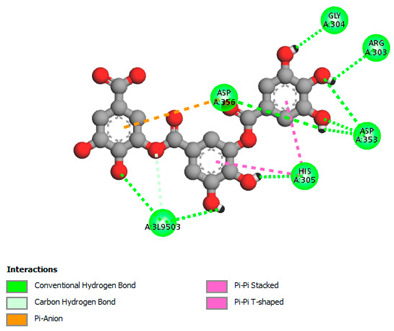	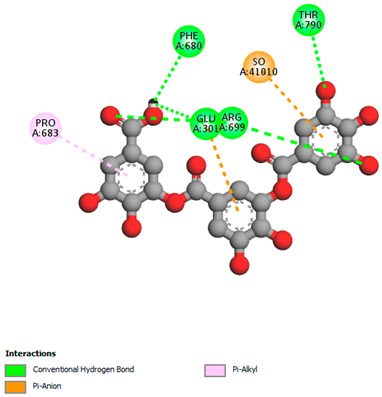
	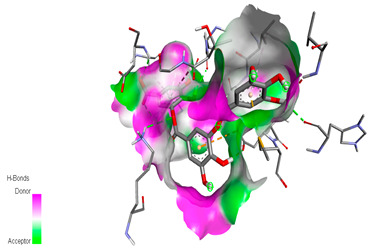	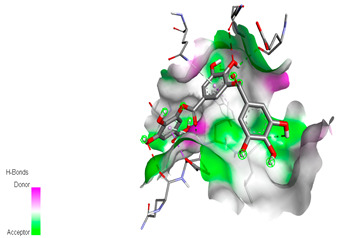	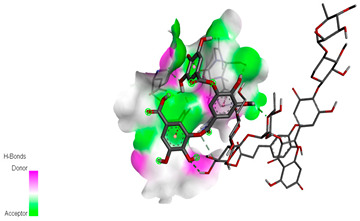	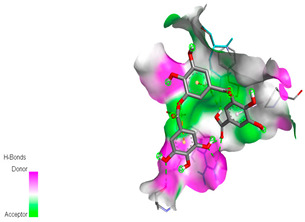
(**B**)	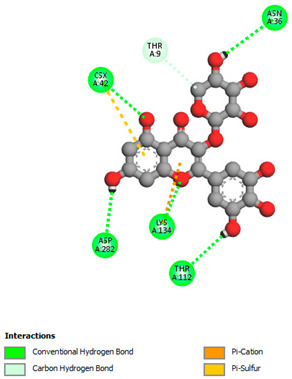	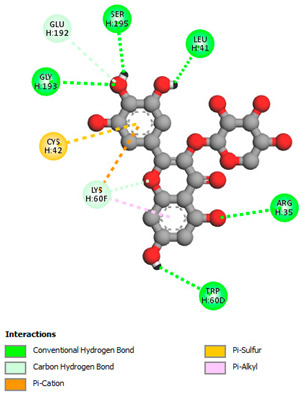	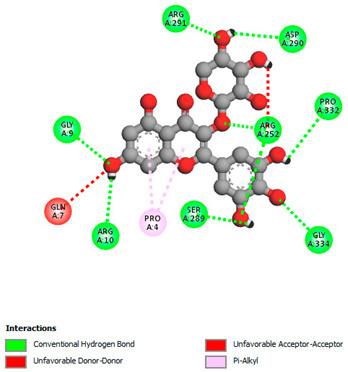	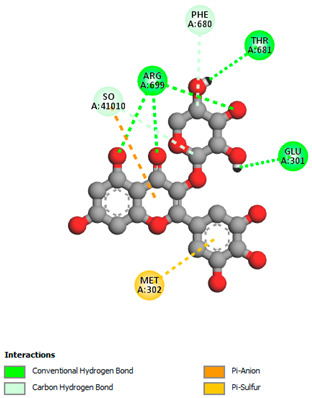
	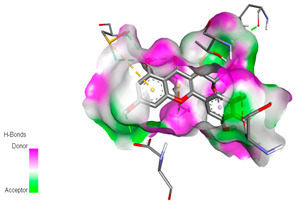	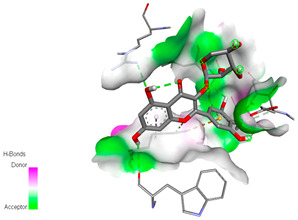	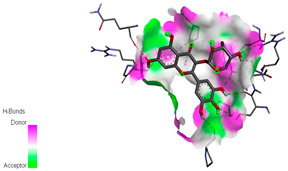	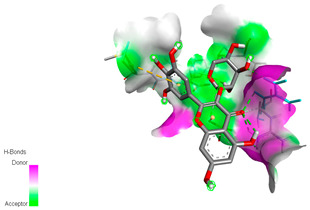
(**C**)	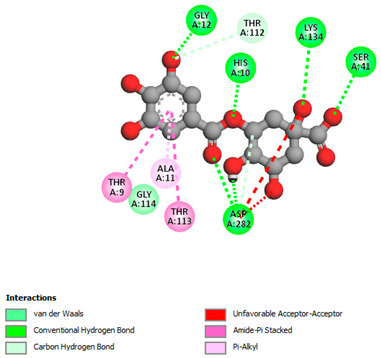	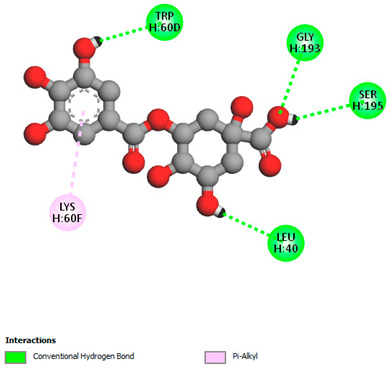	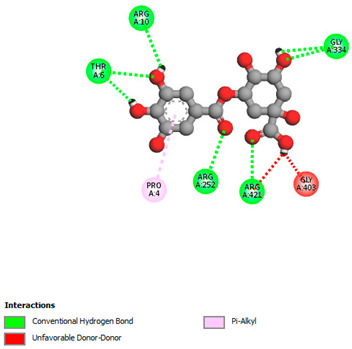	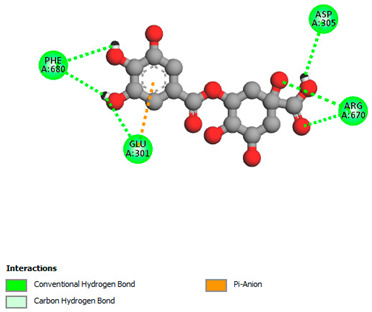
	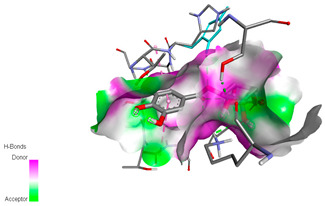	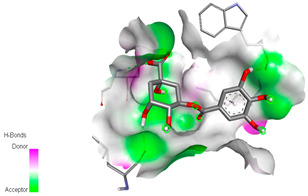	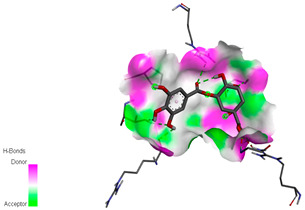	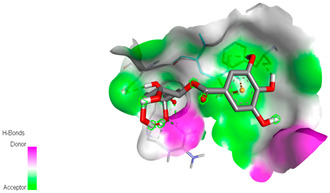
(**D**)	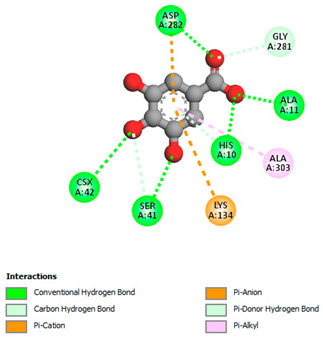	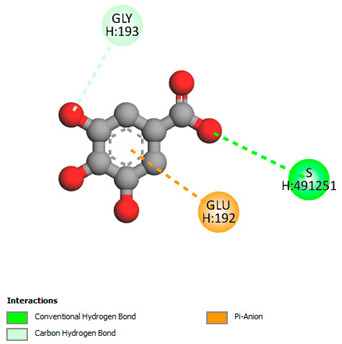	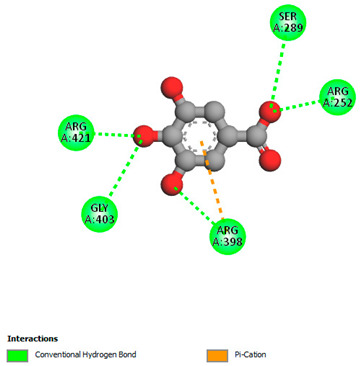	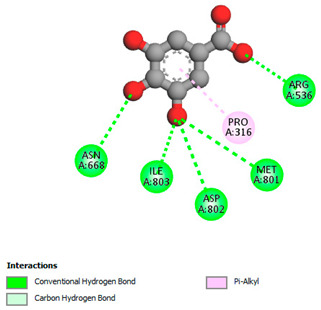
	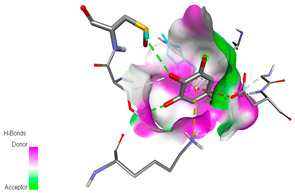	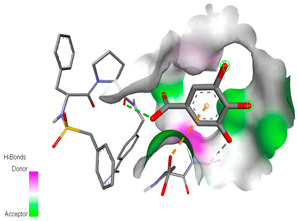	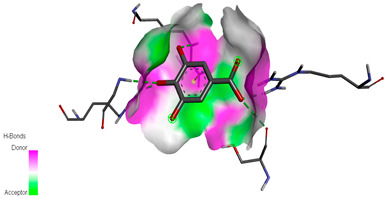	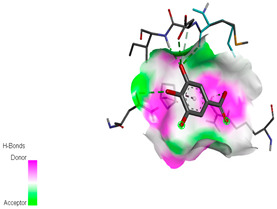
(**E**)	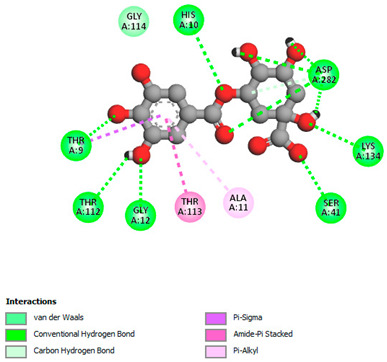	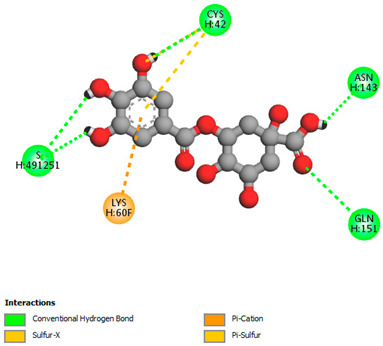	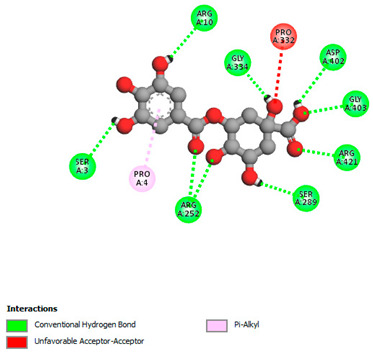	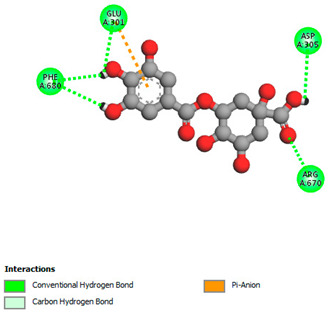
	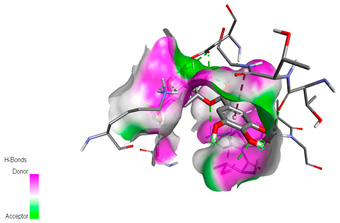	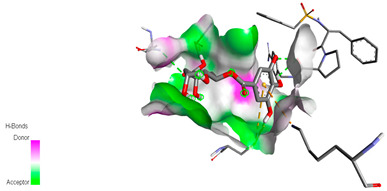	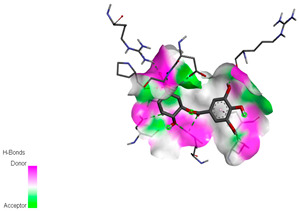	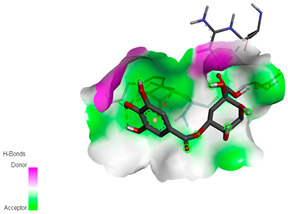
(**F**)	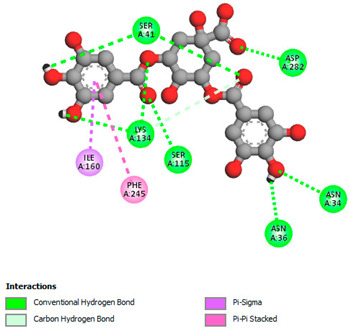	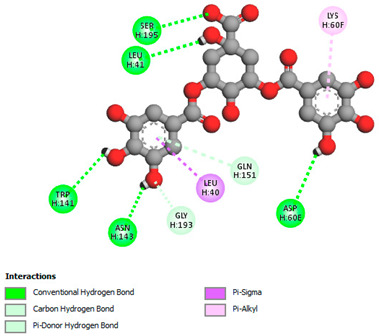	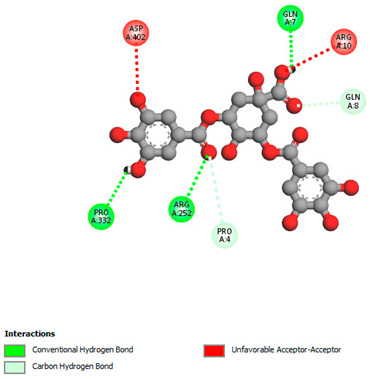	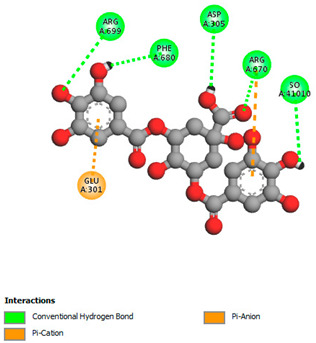
	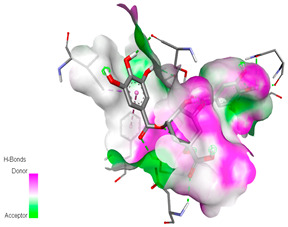	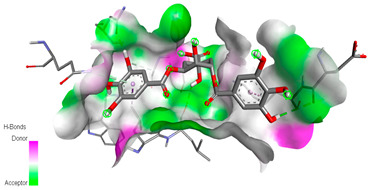	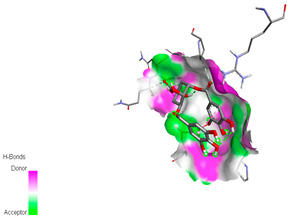	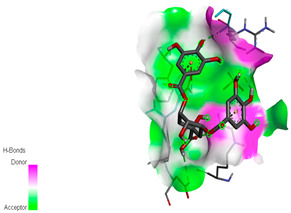
(**G**)	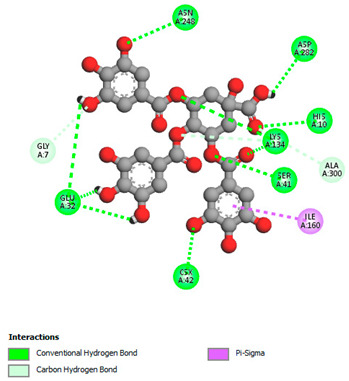	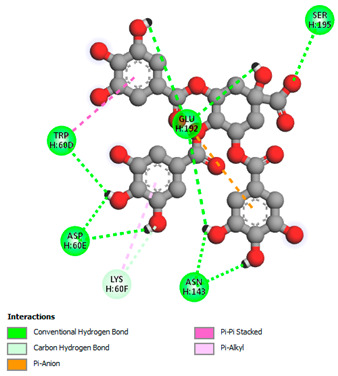	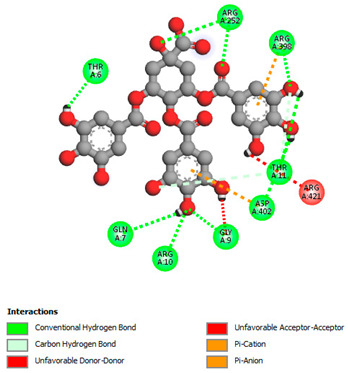	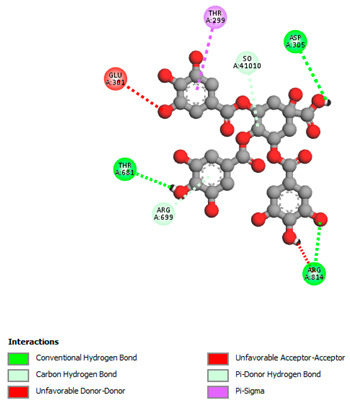
	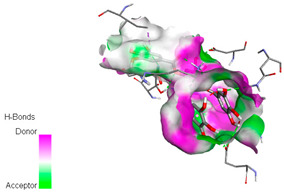	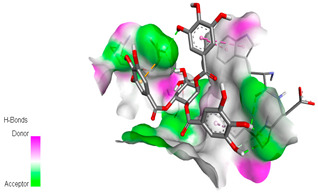	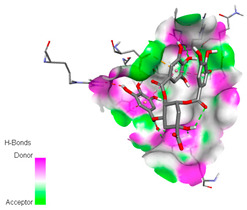	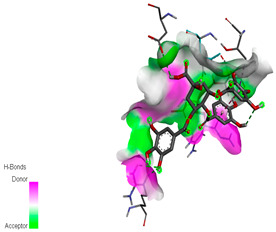

## Data Availability

Not applicable.

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
