# Peer review of "Analysis of the Chemical Composition and Evaluation of the Antioxidant, Antimicrobial, Anticoagulant, and Antidiabetic Properties of Pistacia lentiscus from Boulemane as a Natural Nutraceutical Preservative"

_biomedicines, 2023, doi:10.3390/biomedicines11092372_

Round 1

Reviewer 1 Report

The title of manuscript is remarkable. English language has good quality. Figures and tables have acceptable quality. There are some explainations that are needed about the section "Materials and methods" and "Discussion"

1. About section "2.1. Vegetal Material " in page 3

Please mention the name of the scientists that have performed species' identification

2. About the section "Discussion"

+ please explain about drugs that already are available in the market or drugs that are undergone clinical trials with the similar origin with compounds in this manuscript

+ As you have assessed the bioavailability assay of P.lentiscus extract based

components, it is recommend that you perform prediction of their activity and also their toxicity effects as future drug by some online tools including ProTox-II and Way2Drug

+ you have performed molecular docking. Please draw a figure to show molecular pathways based on you docking findings

+ the authors should write about the safety of P.lentiscus extract based components with other studies specially clinical trials and current common available drugs that are being used in order to cure bacterial and fungul infection, as well as anti-coagulat therapy

3. Please check and adjust the "Reference list" based on the regulations of reference list of journal. (Titles, doi, the name of journal and ... )

Author Response

Dear Professor,

Firstly, I would like to thank you for your valuable suggestions on how to improve this manuscript.

Regarding the identification of the Pistacia lentiscus species, I have made the necessary corrections in the plant material section (page 3, lines 131-132).

Furthermore, I have added the necessary explanations for the Discussion section, which are highlighted in blue.

Regarding your comment on performing molecular docking and drawing a figure to show molecular pathways based on the findings, I am not quite clear on what you are requesting. Would you like a figure of the protein-ligand complex from the docking or a figure that summarizes the in silico study (docking) in relation to the mechanism of action of the studied molecules with respect to the studied pathologies ?

Lastly, I have revised the "Reference List" according to the regulations of the journal's reference list.

Best regards,

Cordialement

Reviewer 2 Report

The article is scientifically relevant as it investigates the chemical composition and evaluates the antioxidant, antimicrobial, anticoagulant, and antidiabetic properties of Pistacia lentiscus L. The study demonstrates the plant's rich phenolic content, strong antioxidant activity, and effective antimicrobial and anticoagulant properties. Additionally, the research includes in-silico analysis to support the in vitro and in vivo findings. These results suggest the potential of Pistacia lentiscus as a natural preservative in the pharmaceutical and agro-food industries.

Introduction

Overall, the introduction provides a clear background and rationale for the study, introducing the topic, the plant species of interest, and the objectives of the research. The introduction appears to be well-structured and informative. However, to enhance it further, you could consider adding the following points:

1.        context: Briefly mention the global prevalence of food preservatives and the potential health concerns associated with synthetic chemical preservatives to provide a broader perspective.

2.       Research objectives: Explicitly state the main objectives of the study, which may include analyzing the chemical composition, evaluating specific properties, and exploring potential applications of Pistacia lentiscus as a natural preservative.

Results

1.       Improve the quality of the figures.

2.       put the "O" in the names of compounds in italics, and italicized “m/z”.

3.       Take care that the minus sign in the molecular masses is in superscript. [M-H]”−“

4.       remove vertical lines from table 8 and table 9.

5.       in table 9, better distribution of results to make it clearer. For example, placing control values ​​below the table.

Conclusion

1.       Overall, the discussion is clear and provides valuable insights into the study's findings. However, there are a few points that could be improved or emphasized based on the results already presented:

2.       Highlight the Potential Applications: Emphasize the potential applications of the identified compounds and their properties as therapeutic agents. Discuss how they could be utilized in nutraceutical preservatives and traditional medicine.

3.       Address Variations in EO Composition: Discuss the factors that may contribute to variations in the composition of P. lentiscus EO observed in different studies, such as plant age, location, and harvesting time.

4.       Correlate Antioxidant Activity and Phenolic Compounds: Discuss the relationship between the high antioxidant activity of P. lentiscus extracts and the presence of phenolic compounds, especially gallic acids and their derivatives.

5.       Discuss Antidiabetic Mechanisms: Explore the potential mechanisms by which the identified compounds in P. lentiscus EO and aqueous extract exhibit antidiabetic effects, particularly their inhibition of α-amylase and α-glucosidase.

6.       Consider Biomedical Relevance: Relate the findings to the potential biomedical relevance of P. lentiscus extracts, especially their antibacterial and antithrombotic properties, which may have implications in various disease treatments.

7.       Stress Pharmacokinetic Properties: Highlight the favorable pharmacokinetic properties predicted by PASS and ADMET analyses, which suggest the compounds' potential as therapeutic agents.

Author Response

Dear Professor,

Firstly, I would like to thank you for your valuable suggestions on how to improve this manuscript.

Regarding your comments on the discussion section, context, study objectives, italicized terms, and table adjustments, they have all been corrected precisely in accordance with your directions. The changes have been highlighted in blue.

Lastly, I have revised the "Reference List" according to the regulations of the journal's reference list.

Best regards,

Cordialement

Round 2

Reviewer 1 Report

I do not have more suggestions.